# AUDITING PREDICTIVE MODELS FOR INTERSECTIONAL BIASES

## ABSTRACT

Predictive models that satisfy group fairness criteria in aggregate for members of a protected class, but do not guarantee subgroup fairness, could produce biased predictions for individuals at the intersection of two or more protected classes. To address this risk, we propose Conditional Bias Scan (CBS), an auditing framework for detecting intersectional biases in classification models. CBS identifies the subgroup with the most significant bias against the protected class, compared to the equivalent subgroup in the non-protected class, and can incorporate multiple commonly used fairness definitions for both probabilistic and binarized predictions. We show that this methodology can detect subgroup biases in the COMPAS pretrial risk assessment tool and in German Credit Data, and has higher bias detection power compared to similar methods that audit for subgroup fairness.

## 1 INTRODUCTION

Predictive models are increasingly used to assist in high-stakes decisions with significant impacts on individuals' lives and livelihoods. However, recent studies have revealed numerous models whose predictions contain biases, in the form of group fairness violations, against disadvantaged and marginalized groups (Angwin et al., 2016a; Obermeyer et al., 2019). When auditing a predictive model for bias, typical group fairness definitions (Mitchell et al., 2021) rely on univariate measurements of the difference between the distributions of predictions or outcomes for individuals in a "protected class", typically defined by a sensitive attribute such as race or gender, as compared to those in the non-protected class. Since these approaches only detect biases for a predetermined subpopulation at an aggregate level, e.g., a bias against Black individuals, they may fail to detect biases that adversely affect a subset of individuals in a protected class, e.g., Black females. While it is possible to define a specific multidimensional subgroup and then audit a classifier for biases impacting that subgroup, this approach does not scale to the combinatorial number of subgroups. Therefore, group fairness measurements cannot reliably detect if there are *any* subgroups within a given population that are adversely impacted by predictive biases, and thus subgroup biases in predictions often go unaddressed.

In this paper, we present a novel methodology for bias detection called Conditional Bias Scan (CBS). Given a classifier's probabilistic *predictions* or binarized *recommendations* based on those predictions, CBS discovers systematic biases impacting any *subgroups* of a predefined subpopulation of interest (the *protected class*). More precisely, CBS aims to discover subgroups of the protected class for whom the classifier's predictions or recommendations systematically deviate from the corresponding subgroup of individuals who are not a part of the protected class. Subgroups are defined by a non-empty subset of attribute values for each observed attribute, excluding the *sensitive attribute* which determines whether or not individuals belong to the protected class.

The detected subgroups can represent both *intersectional* and *contextual* biases. *Intersectional* biases refer to subgroup biases defined by membership in two or more protected classes. See Appendix D and references (Crenshaw, 1991a; Runyan, 2018) for further discussion of the concept of intersectionality. *Contextual* biases refer to other forms of subgroup biases that may only be present for certain decision situations (Runyan, 2018). For example, when auditing an algorithmic risk assessment tool, CBS may identify a subgroup bias against Black females (intersectional bias) for individuals with no prior offenses (contextual bias).

Table 1: Table of all scan types for CBS for different group fairness definitions.

| | | Predictions ($P \in [0,1]$) | Recommendations ($P_{bin} \in \{0,1\}$) | | |
| --- | --- | --- | --- | --- | --- |
| | | | $P_{bin} = 1$ | $P_{bin} = 0$ | $P_{bin}$ |
| Separation | $Y = 1$ | $\mathbb{E}[P \mid Y=1,X] \perp A$ 
 *Balance for Positive Class* | $\Pr(P_{bin}=1 \mid Y=1,X) \perp A$ 
 *True Positive Rate* | $\Pr(P_{bin}=0 \mid Y=1,X) \perp A$ 
 *False Negative Rate* | |
| | $Y = 0$ | $\mathbb{E}[P \mid Y=0,X] \perp A$ 
 *Balance for Negative Class* | $\Pr(P_{bin}=1 \mid Y=0,X) \perp A$ 
 *False Positive Rate* | $\Pr(P_{bin}=0 \mid Y=0,X) \perp A$ 
 *True Negative Rate* | |
| | $Y$ | $\mathbb{E}[P \mid Y,X] \perp A$ | $\Pr(P_{bin}=1 \mid Y,X) \perp A$ | $\Pr(P_{bin}=0 \mid Y,X) \perp A$ | |
| Sufficiency | $Y = 1$ | $\Pr(Y=1 \mid P,X) \perp A$ | $\Pr(Y=1 \mid P_{bin}=1,X) \perp A$ 
 *Positive Predictive Value* | $\Pr(Y=1 \mid P_{bin}=0,X) \perp A$ 
 *False Omission Rate* | $\Pr(Y=1 \mid P_{bin},X) \perp A$ |
| | $Y = 0$ | $\Pr(Y=0 \mid P,X) \perp A$ | $\Pr(Y=0 \mid P_{bin}=1,X) \perp A$ 
 *False Discovery Rate* | $\Pr(Y=0 \mid P_{bin}=0,X) \perp A$ 
 *Negative Predictive Value* | $\Pr(Y=0 \mid P_{bin},X) \perp A$ |

The notation $\perp$ refers to conditional independence from membership in the protected class ($A$). For example, for the False Discovery Rate scan, $\Pr(Y = 0 \mid P_{bin} = 1, X) \perp A$ is shorthand for $\Pr(Y = 0 \mid P_{bin} = 1, X, A = 1) = \Pr(Y = 0 \mid P_{bin} = 1, X, A = 0)$.

The contributions of our research include:

- A methodological framework that can flexibly accommodate multiple group-fairness definitions and can reliably detect intersectional and contextual biases, with significantly improved bias detection accuracy compared to other tools used to audit for subgroup fairness.

- A computationally efficient detection algorithm to audit classifiers for fairness violations in the exponentially many subgroups of a prespecified protected class.

- Robust evaluation and two real-world case studies that compare results across group-fairness metrics, showing differences between separation and sufficiency metrics.

## 2 RELATED WORK

Bias Scan (Zhang and Neill, 2016) uses a multidimensional subset scan to search exponentially many subgroups of data, identifying the subgroup with the most significantly miscalibrated probabilistic predictions compared to the observed outcomes. Bias Scan lacks the functionality of traditional group fairness techniques to define a protected class and to determine whether those individuals are impacted by biased predictions, and is thus limited to asking, "Which subgroup has the most miscalibrated predictions?" In contrast, given a protected class $A$, CBS can reliably identify biases impacting $A$ or any subgroup of $A$. CBS searches for subgroups within the protected class with the most significant deviation in their predictions and observed outcomes as compared to the predictions and observed outcomes for the corresponding subgroup of the non-protected class (e.g., a racial bias against Black females as compared to non-Black females). Since Bias Scan solely focuses on the deviation between the predictions and observed outcomes within a subgroup, it would be unable to detect such a bias unless the subgroup was also biased as compared to the population as a whole. Furthermore, CBS generalizes to separation- and sufficiency-based group fairness metrics, and to probabilistic and binarized predictions. To enable this new functionality, CBS deviates from Bias Scan in substantial ways, including new preprocessing and estimation techniques (see Section 3.2 and Appendix A.1) and new hypotheses and score functions (see Section 3.3).

GerryFair (Kearns et al., 2018) and MultiAccuracy Boost (Kim et al., 2019a) are two methods that use an auditor to iteratively detect subgroups while training or correcting a classifier to guarantee subgroup fairness. GerryFair's auditor relies on linear regressions trained to predict differences between the predictions and the observed global error rate of a dataset. MultiAccuracy Boost iteratively forms subgroups by evaluating rows with predictions above and below a threshold to determine which predictions to adjust. CBS's methodology for forming subgroups is more complex because it does not assume a linear relationship between covariates and the difference between the predictions and baseline error rate. Unlike CBS, these methods provide limited fairness definitions for auditing, and do not return interpretable subgroups that are defined by discrete attribute values of the covariates, but rather identify all rows that have a fairness violation on a given iteration. Since both methods incorporate the predictions in forming subgroups and enable auditing, they are comparable to CBS. In Section 4, we show that CBS has substantially higher bias detection accuracy than GerryFair and

MultiAccuracy Boost. Additional related work about subgroup bias, intersectionality, and subgroup discovery is discussed in Appendix D.

# 3 METHODS

CBS begins by defining the dataset $D = (A, X, Y, P, P_{bin}) = \{(A_i, X_i, Y_i, P_i, P_{i,bin})\}_{i=1}^{n}$, for $n$ individuals indexed as $i = 1..n$. $A_i$ is a binary variable representing whether individual $i$ belongs to the protected class. $X_i = (X_i^1 \ldots X_i^m)$ are other covariates for individual $i$, excluding $A_i$ and the sensitive attribute from which $A_i$ was derived. We assume here that all covariates are discrete-valued, but continuous covariates can also be used (see Appendix A.1 for discussion). $Y_i$ is individual $i$'s observed binary outcome, $P_i \in [0, 1]$ is the classifier's probabilistic prediction of individual $i$'s outcome, and $P_{i,bin} \in \{0, 1\}$ is the binary recommendation corresponding to $P_i$.

Given these data, CBS searches for subgroups of the protected class, defined by a non-empty subset of values for each covariate $X^1 \ldots X^m$, for whom some *group fairness definition* (contained in Table 1) is violated. Each fairness definition can be viewed as a conditional independence relationship between an individual's membership in the protected class $A_i$ and their value of an *event variable* $I_i$, conditioned on their covariates $X_i$ and their value of a *conditional variable* $C_i$. We define the null hypothesis, $H_0$, that $I \perp A \mid (C, X)$, and use CBS to search for subgroups with statistically significant violations of this conditional independence relationship, correctly adjusting for multiple hypothesis testing, allowing us to reject $H_0$ for the alternative hypothesis $H_1$ that $I \not\perp A \mid (C, X)$.

The CBS framework has four sequential steps. (1) Given a fairness definition, CBS chooses $I \in \{Y, P, P_{bin}\}$ and $C \in \{Y, P, P_{bin}\}$. Section 3.1 maps different group fairness criteria to particular choices of event variable $I$ and conditional variable $C$. (2) CBS estimates the expected value of $I_i$ for each individual in the protected class under the null hypothesis $H_0$ that $I$ and $A$ are conditionally independent. These expectations are denoted as $\hat{I}_i$. Section 3.2 describes how to estimate $\hat{I}$. (3) CBS uses a novel *multidimensional subset scan* to search for subgroups $S$ where for $i \in S$, $I_i$ deviates systematically from its expectation $\hat{I}_i$ in the direction of interest. This step to *detect* $S^*$ is described in Section 3.3. (4) The final step to *evaluate statistical significance* of the detected subgroup $S^*$ (Section 3.3) uses permutation testing (Appendix A.3) to adjust for multiple hypothesis testing and determine if $S^*$'s deviation between protected and non-protected class is statistically significant.

## 3.1 DEFINE $(I, C)$: *Overview of Scan Types*

Many of the group fairness criteria proposed in the fairness literature fall into two categories of statistical fairness called sufficiency and separation. *Sufficiency* is focused on equivalency in the rate of an outcome (for comparable individuals with the same prediction or recommendation) regardless of protected class membership ($Y \perp A \mid P, X$), whereas *separation* is focused on equivalency of the expected prediction or recommendation (for comparable individuals with the same outcome) regardless of protected class membership ($P \perp A \mid Y, X$). The choice between separation and sufficiency determines whether outcome $Y$ is the event variable of interest $I$ or the conditional variable $C$, where bias exists if $\mathbb{E}[I \mid C, X, A = 1] \neq \mathbb{E}[I \mid C, X, A = 0]$. The combination of fairness metric (sufficiency or separation) and prediction type (continuous prediction or binary recommendation) produces four classes of fairness scans: separation for predictions ($I = P, C = Y$), separation for recommendations ($I = P_{bin}, C = Y$), sufficiency for predictions ($I = Y, C = P$), and sufficiency for recommendations ($I = Y, C = P_{bin}$).

Depending on the particular bias of interest, we can also perform "value-conditional" scans by restricting the value of the conditional variable. For example, to scan for subgroups with increased false positive rate (FPR), we restrict the data to individuals with $Y = 0$ and perform a separation scan for recommendations. All of the scan options for CBS are shown in Table 1. Each scan in Table 1 can detect bias in either direction, e.g., searching for subgroups with increased or decreased FPR.

## 3.2 GENERATE EXPECTATIONS $\hat{I}$ OF THE EVENT VARIABLE

Once we have defined the event variable $I$ and conditional variable $C$, we wish to detect fairness violations by assessing whether there exist subgroups of the protected class where $\mathbb{E}[I \mid C, X, A = 1]$ differs systematically from $\mathbb{E}[I \mid C, X, A = 0]$. For each individual $i$ in the protected class,

Table 2: Null and alternative hypotheses, $H_0$ and $H_1(S)$, and corresponding log-likelihood ratio score functions, $F(S)$, used to measure a subgroup's degree of anomalousness (comparing the event variable $I$ to its expectation $\hat{I}$ under $H_0$) for all four variants of CBS.

| Scan Types | | Hypotheses | Distribution for $F(S)$ | $F(S)$ |
|---|---|---|---|---|
| Separation | Predictions | Null Hypothesis: $H_0: \Delta_i \sim N(0,\sigma), \forall i \in D_1$ 
 Alternative Hypothesis: $H_1(S): \Delta_i \sim N(\mu,\sigma)$ 
 where $\Delta_i = \log\left(\frac{I_i}{1-I_i}\right) - \log\left(\frac{\hat{I}_i}{1-\hat{I}_i}\right)$ 
 Over-estimation Bias: $\mu < 0, \forall i \in S$, and $\mu = 0, \forall i \notin S$. 
 Under-estimation Bias: $\mu > 0, \forall i \in S$, and $\mu = 0, \forall i \notin S$. | Gaussian | $\max_\mu \frac{2\mu\left(\sum_{i\in S}\Delta_i\right)-|S|\mu^2}{2\sigma^2}$ |
| Sufficiency | Recommendations | Null Hypothesis: $H_0: odds(I_i) = \frac{I_i}{1-I_i}, \forall i \in D_1$ | | |
| | Predictions | Alternative Hypothesis: $H_1(S): odds(I_i) = q\frac{\hat{I}_i}{1-\hat{I}_i}$ | Bernoulli | $\max_q \sum_{i\in S}(I_i\log(q)$ |
| | Recommendations | Over-estimation Bias: $q < 1, \forall i \in S$, and $q = 1, \forall i \notin S$. 
 Under-estimation Bias: $q > 1, \forall i \in S$, and $q = 1, \forall i \notin S$. | | $-\log(q\hat{I}_i - \hat{I}_i + 1))$ |

Over-estimation (under-estimation) bias means that the expectations $\hat{I}_i$ are larger (smaller) than $I_i$.

$I_i \mid C_i, X_i, A_i = 1$ is observed but $I_i \mid C_i, X_i, A_i = 0$ is unobserved. Thus we must calculate an estimate $\hat{I}_i = \mathbb{E}_{H_0}[I_i \mid C_i, X_i, A_i = 1]$, under the null hypothesis, $H_0: (I \perp A \mid C, X)$, and compare $\hat{I}_i$ to the observed $I_i$. To calculate $\hat{I}$ we use the following method from the econometric literature on heterogeneous treatment effects, which controls for non-random selection into the protected class $A$ based on observed covariates $X$: (1) Learn a probabilistic model for estimating $\Pr(A = 1 \mid X)$, and use it to produce propensity scores, $p_j^A$, for each individual $j$ in the non-protected class; (2) For each individual $j$ in the non-protected class, use the observed $\mathbb{E}[I_j \mid C_j, X_j, A_j = 0]$ weighted by the odds of the propensity score for individual $j$, $\frac{p_j^A}{1-p_j^A}$, to learn a probabilistic model for $\mathbb{E}_{H_0}[I \mid C, X, A = 1]$; (3) For each individual $i$ in the protected class, use the model of $\mathbb{E}_{H_0}[I \mid C, X, A = 1]$ to calculate $\hat{I}_i = \mathbb{E}_{H_0}[I_i = 1 \mid C_i, X_i, A_i = 1]$. Appendix A.1 provides a detailed description of this method, including its modifications for a real-valued event variable (i.e., separation scan for predictions) and for value-conditional scans.

## 3.3 DETECT THE MOST SIGNIFICANT SUBGROUP $S^*$ AND EVALUATE ITS STATISTICAL SIGNIFICANCE

Given the observed event variables $I_i$ and the expectations $\hat{I}_i$ of the event variable under the null hypothesis ($I \perp A \mid C, X$) for the protected class, we define a score function measuring *subgroup bias*, $F : S \to \mathbb{R}_{\geq 0}$, that can be efficiently optimized over exponentially many subgroups to identify $S^* = \arg\max_S F(S)$. To do so, we follow the literature on spatial and subset scan statistics (Kulldorff, 1997; Neill, 2012) by defining score functions $F(S)$ that take the general form of a log-likelihood ratio (LLR) test statistic, $F(S) = \log\left(\frac{\Pr(D \mid H_1(S))}{\Pr(D \mid H_0)}\right)$. Here the denominator represents the likelihood of seeing the observed values of event variable $I$ for subgroup $S$ of the protected class under the null hypothesis $H_0$ of no bias. The numerator represents the likelihood of seeing the observed values of $I$ for subgroup $S$ of the protected class under the alternative hypothesis $H_1(S)$, where the $I_i$ values are systematically increased or decreased as compared to $\hat{I}_i$. For $H_1(S)$ to represent a deviation from $H_0$, $H_1$ contains a free parameter ($q$ or $\mu$) that is determined by maximum likelihood estimation. Under-estimation bias ($I_i > \hat{I}_i$) or over-estimation bias ($I_i < \hat{I}_i$) can be detected using different constraints for $q$ or $\mu$ as shown in Table 2. When $I$ is a probabilistic prediction (i.e., for separation scan for predictions), the hypotheses are in the form of a difference of log-odds between $I$ and $\hat{I}$ sampled from a Gaussian distribution. Here the free parameter $\mu$ in $H_1$ represents a mean shift ($\mu \neq 0$) of the Gaussian distribution. For all other scans, under $H_0$, each observed $I_i$ is assumed to be drawn from a Bernoulli distribution centered at the corresponding expectation $\hat{I}_i$. Under $H_1$, the free parameter $q$ represents a multiplicative increase or decrease ($q \neq 1$) of the odds of $I$ as compared to $\hat{I}$. The various score functions all aggregate the deviations from $H_0$ for each instance in a subgroup, and thus the log-likelihood ratio score $F(S)$ scales linearly with subgroup size $|S|$ for a given amount of deviation. This dependence on $|S|$ prevents the scan from assigning disproportionately high log-likelihood scores to subgroups with very few instances where there is a

large deviation in, for example, false positive rates between those in the protected class and those in the non-protected class. This helps to ensure that subgroups with few instances with large, chance deviations from the null hypothesis are not favored over the true, larger subgroups of interest.

As in Zhang and Neill (2016), a penalty term can be added to $F(S)$ equal to a prespecified scalar times the total number of attribute values included in subgroup $S$, summed across all covariates $X^1 \ldots X^m$. Note that there is no penalty for a given attribute if all attribute values are included, since this is equivalent to ignoring the attribute when defining subgroup $S$. The penalty term results in more interpretable subgroups by encouraging the scan to either ignore an attribute (i.e., all values of that attribute are included in the subgroup) or choose a smaller number of attribute values to include in the subgroup. This allows the detected subgroup to consist of those attributes and values whose inclusion most increases the log-likelihood ratio score, while omitting those attributes and values that have little effect on the log-likelihood ratio score.

We now consider how CBS is able to efficiently maximize $F(S)$ over subgroups $S$ of the protected class, returning $S^* = \arg\max_S F(S)$ and the corresponding score $F(S^*)$. The scan procedure for CBS takes as inputs a dataset $D_1 = (I, \hat{I}, X)$ consisting of the event variable $I_i$, the estimated expectation of $I_i$ under the null hypothesis $\hat{I}_i$, and the covariates $X_i$, for each individual in the protected class ($A_i = 1$), along with several parameters: the type of scan (Gaussian or Bernoulli), the direction of bias to scan for (over- or under-estimation), complexity penalty, and number of iterations. It then searches for the highest-scoring subgroup (consisting of a non-empty subset of values $V^j$ for each covariate $X^j$), starting with a random initialization on each iteration, and proceeding by *coordinate ascent*. The coordinate ascent step identifies the highest-scoring non-empty subset of values $V^j$ for a given covariate $X^j$, conditioned on the current subsets of values $V^{-j}$ for all other attributes. As shown in McFowland III et al. (2023), each individual coordinate ascent step can provably find the optimal subset of attribute values while evaluating only $|X^j|$ of the $2^{|X^j|}$ subsets of values, where $|X^j|$ is the arity of covariate $X^j$. This efficient subroutine follows from the fact that the score functions above satisfy the additive linear-time subset scanning property (Neill, 2012; Speakman et al., 2016). The coordinate ascent step is repeated with different, randomly selected covariates until convergence to a local optimum of the score function, and multiple random restarts enable the scan to approach the global optimum. McFowland III et al. (2023) provide sufficient conditions under which this routine will identify the global optimum in the large-sample limit; empirically, the approach converges to near-optimal subgroups while requiring only low-order polynomial time. For an in-depth, self-contained description of the scan algorithm, including pseudocode, and how it exploits an additive property of the score functions to achieve linear-time efficiency for each scan step, see Appendix A.2. Finally, as described in detail in Appendix A.3, we perform *permutation testing* to compute the p-value of the detected subgroup, comparing its score to the distribution of maximum subgroup scores under $H_0$, and report whether it is significant at a given level $\alpha$ (e.g., $\alpha = .05$).

## 4 EVALUATION

Given the lack of gold standard approaches for evaluating subgroup bias auditing methods, we evaluate the CBS framework through semi-synthetic simulations with the following steps:

(A) Randomly select a protected class $A$ and *generate a semi-synthetic dataset* where the predictions, recommendations, and outcomes are conditionally independent of $A$ given $X$, i.e., there are *no sufficiency or separation violations* (as defined in Section 3.1) pertaining to protected class $A$.

(B) Take the unmodified semi-synthetic data and *inject signal* consistent with a separation or sufficiency violation or base rate shift into a subgroup of protected class $A$.

(C) *Run CBS and benchmark methods* to detect violations pertaining to protected class $A$ and *measure the accuracy of the detected subgroups* compared to the known (injected) biased subgroup.

We generate 100 semi-synthetic datasets. For each dataset, we perform the same set of 1,344 experiments, each with a specific type and amount of injected signal. We then average performance over the 100 datasets for each experiment.

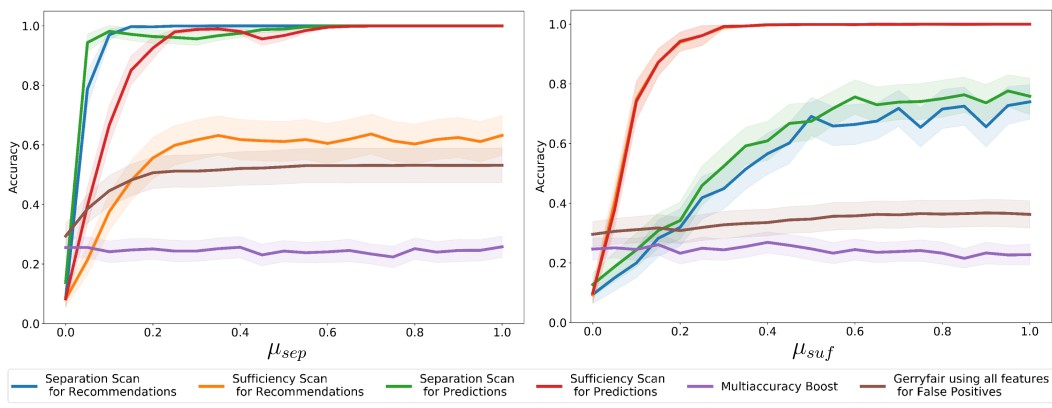

Figure 1: Average accuracy (with 95% CI) as a function of the amount of bias injected into subgroup $S_{bias}$ of the protected class, for four variants of CBS, GerryFair, and MultiAccuracy Boost. Left: increasing predicted probabilities by $\mu_{sep}$. Right: decreasing true probabilities by $\mu_{suf}$.

*(A) Generate a semi-synthetic dataset:* Using COMPAS data[1] described in Section 5, we randomly select an attribute and value to define the protected class $A$ and remove that attribute from $X$. For each attribute-value of the covariates, we draw a weight from a Gaussian distribution, $\mathcal{N}(0, 0.2)$. We use these weights to produce the true log-odds of a positive outcome ($Y_i = 1$) for each row $i$ by a linear combination of the attribute values with these weights. Additionally, for each row, we add $\epsilon_i^{true} \sim \mathcal{N}(0, \sigma_{true})$ to its true log-odds, representing variation between rows that arises from external factors (not included in the scan attributes), and is incorporated into the predictive model.[2] Given the true log-odds $L_i^{true}$ of $Y_i = 1$ for each row, we draw each outcome $Y_i$ from a Bernoulli distribution with the corresponding probability, expit($L_i^{true}$), which we refer to as the true probabilities. Next, we set each row's predicted probability $P_i = $ expit($L_i^{true} + \epsilon_i$), where $\epsilon_i \sim \mathcal{N}(0, \sigma_{predict})$ represents non-systematic errors (random noise) in the predictive model. We use default values of $\sigma_{true} = 0.6$ and $\sigma_{predict} = 0.2$, and examine sensitivity to these parameters in Appendix B.4; see Appendix B.2 for discussion of the impact of $\sigma_{true}$ on sufficiency-based fairness definitions. Finally, we threshold the probabilities to produce recommendations $P_{i,bin} = \mathbf{1}(P_i \geq 0.5)$ for each row $i$. Since $A$ is conditionally independent of the outcomes $Y$, predictions $P$ and recommendations $P_{bin}$ given the observed covariates $X$, this dataset contains *no* signals indicating separation or sufficiency violations for a subgroup of protected class $A$.

*(B) Inject signal:* We randomly select a subgroup of the protected class $S_{bias}$ into which we will inject biases or base rate shifts. We pick $S_{bias}$ by randomly choosing two attributes ($n_{bias} = 2$) and then independently including or excluding each value of those attributes with probability $p_{bias} = 0.5$. (This process is repeated until the resulting subgroup is non-empty.)

We designed the evaluation to address three key questions about the performance of the four CBS variants and benchmark methods:

(Q1) How well do they detect *biases* represented as systematic differences between the predicted and true probabilities for the event variable $I$ in subgroup $S_{bias}$ of the protected class?

(Q2) How do they respond to a *base rate shift*, i.e., an equal shift $\delta$ in the predicted and true probabilities for the event variable $I$ for subgroup $S_{bias}$ of the protected class, assuming no injected bias?

(Q3) How do the answers to the first two questions vary based on the characteristics of $S_{bias}$?

---

[1] We use the covariates from COMPAS to maintain realistic covariate correlations, but do not use the predictions or outcomes.

[2] Rudin et al. (2020) note that COMPAS relies on up to 137 variables collected from a questionnaire, and we expect that some of these additional variables are correlated with outcomes.

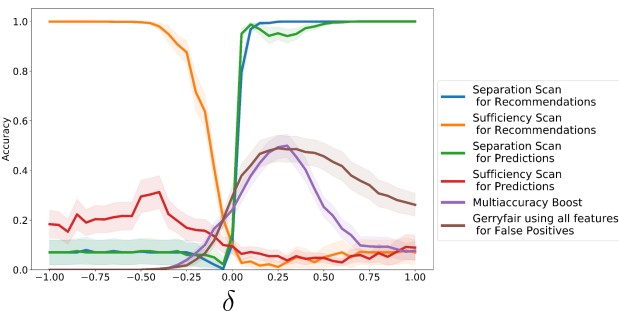

Figure 2: Average accuracy (with 95% CI) as a function of the base rate difference $\delta$ between protected and non-protected class for subgroup $S_{bias}$, for four variants of CBS, GerryFair, and MultiAccuracy Boost. Note that predictions are well calibrated, $\mu_{sep} = \mu_{suf} = 0$.

To address (Q1), we inject bias into subgroup $S_{bias}$ of the protected class, keeping the corresponding subgroup of the non-protected class unchanged, in one of two ways: (1) increasing the predicted probabilities, $P_i$, by $\mu_{sep}$ for each row in $S_{bias}$, and recomputing the model's recommendations $P_{i,bin}$ by thresholding $P_i$ at 0.50; or (2) reducing the true probabilities by $\mu_{suf}$ for each row in $S_{bias}$, and redrawing the outcomes $Y_i$. Both of these shifts result in a bias where $P$ and $P_{bin}$ overestimate the outcomes ($Y$) for the given subgroup of the protected class. When $\mu_{sep} > 0$, this creates a signal which is consistent with separation violations in the positive direction. When $\mu_{suf} > 0$, this creates a signal which is consistent with sufficiency violations in the negative direction. To address (Q2), we inject a base rate shift into subgroup $S_{bias}$ of the protected class, keeping the corresponding subgroup of the non-protected class unchanged by increasing *both* the true probabilities and the predicted probabilities of $S_{bias}$ by $\delta$, then redrawing outcomes $Y_i$ and recomputing recommendations $P_{i,bin}$. For positive $\delta$, this creates a higher base rate of a positive outcome for subgroup $S_{bias}$ of the protected class, as compared to the corresponding subgroup of the non-protected class, while maintaining well-calibrated predictions.

Importantly, the signals for $\mu_{sep}$, $\mu_{suf}$, and $\delta$ are created by a uniform shift in the true and predicted probabilities, which corresponds to a *non-uniform* shift in the true and predicted log-odds. **This is distinct from the modeling assumption made by CBS**, which assumes (under the alternative hypothesis that bias is present) a constant additive shift in the true or predicted log-odds. By injecting signal in this way, we ensure that our method is robust to non-additive shifts in log-odds. For simulation results that inject bias represented as additive shifts in log-odds, please see Appendix B.4. We observe high consistency between those additional results and the ones presented here.

To address (Q3), we vary the size of $S_{bias}$ by (1) varying the number of attributes, $n_{bias}$, that the attribute-values can be chosen from, between 1 and 4; or (2) varying the probability, $p_{bias}$, that each value of the chosen attributes is included in $S_{bias}$. We run three experiments ($\mu_{sep} = 0.50$, $\mu_{suf} = 0.50$, and $\delta = 0.25$) while varying $n_{bias}$ and $p_{bias}$ for each experiment.

*(C) Run CBS and benchmark methods and measure the accuracy of the detected subgroups:* We compare the four variants of CBS to GerryFair (Kearns et al., 2018) and MultiAccuracy Boost (Kim et al., 2019a), described in Section 2. For more information about the methods and modifications we made to both benchmark methods to make them more comparable to CBS for these simulations, see Appendix B.1. We use the same settings for CBS as described in Section 5, with the exception of running all scans with all conditional variable values rather than as value-conditional scans. After injecting bias into or shifting the base rates of $S_{bias}$ in the protected class and running all CBS scans and GerryFair and MultiAccuracy Boost, we measure the accuracy of a detected subset, $S^*$, by $\text{accuracy}(S^*) = \frac{|\,S_{bias} \cap S^*\,|}{|\,S_{bias} \cup S^*\,|}$, the Jaccard similarity between the injected and detected subsets. This accuracy measure penalizes both falsely detected unbiased instances and undetected instances affected by bias, making it appropriate for applications where both types of error should be minimized. Accuracies are averaged over the 100 simulations for each experiment.

**Simulation Results:** In Figure 1, which addresses (Q1), we observe that all four variants of CBS are able to detect the injected bias (for subgroup $S_{bias}$ of the protected class) with higher accuracy than GerryFair or MultiAccuracy Boost. Sufficiency scans had highest accuracy for shifts in true

probabilities ($\mu_{suf}$), and separation scans had highest accuracy for shifts in predicted probabilities ($\mu_{sep}$). Scans for predictions generally outperformed scans for recommendations, due to the loss of information from binarizing the probabilistic predictions. Interestingly, sufficiency scan for predictions (but not for recommendations) converged to perfect accuracy for $\mu_{sep}$, while separation scans did not converge to perfect accuracy for $\mu_{suf}$. Sufficiency scan for predictions is conditioned on a real-valued variable ($P_i$) rather than a binary variable ($P_{i,bin}$ or $Y_i$), allowing more flexible modeling of $\mathbb{E}[Y \mid P, X]$ and thus greater sensitivity to shifts in predicted probabilities.

In Figure 2, which addresses (Q2), shifting the base rate for subgroup $S_{bias}$ of the protected class results in separation scans detecting a base rate shift when $\delta > 0$, while sufficiency scans and competing methods are robust to this shift. This finding aligns with previous research proving that differences in base rates between two populations will result in a higher false positive rate for the population with a higher base rate when using a well-calibrated classifier (Chouldechova, 2017). Interestingly, sufficiency scan for recommendations detects a base rate shift for $\delta \ll 0$. In this case, $\mathbb{E}[Y \mid P_{bin}, X]$ is lower for instances in the protected class than for instances with negative recommendations in the non-protected class. Thus conditioning on the binary indicator $P_{i,bin}$ is not sufficient to capture this decrease in the true probabilities, while conditioning on the real-valued prediction $P_i$ allows sufficiency scan for predictions to extrapolate reasonably well to these cases.

In Figure 4 in Appendix B.3, which addresses (Q3), we see that CBS is robust to increasing the number of affected dimensions $n_{bias}$, with the relative accuracies for scans and competing methods similar to those in Figures 1 and 2. Interestingly, increasing $p_{bias}$ to 1 (meaning that bias is injected into the entire protected class) enables GerryFair to achieve similar accuracy to CBS for $\mu_{sep} = 0.50$, but CBS outperforms GerryFair for smaller, more subtle, subgroup biases. All fixed hyper-parameter choices for these simulations are moderate values which align with non-edge cases. Additional robustness checks for varying hyper-parameter choices for these simulations are described in Appendix B.4. For estimates of compute power needed for the simulations see Appendix B.5.

## 5 CASE STUDY OF COMPAS

The COMPAS algorithm is used in various jurisdictions across the United States as a decision support tool to predict individuals' risk of recidivism. It is commonly used by judges when deciding whether an arrested individual should be released prior to their trial (Angwin et al., 2016b). We define each defendant's predicted probability of reoffending, $P_i$, by mapping their COMPAS risk score to the proportion of all defendants with the given risk score who reoffended. Defendants with COMPAS risk scores of 5+ are considered "high risk" ($P_{i,bin} = 1$) since the COMPAS documentation stipulates careful consideration by supervision agencies for these defendants (Larson et al., 2016). For details about the COMPAS data, critiques of this dataset, and other considerations about using COMPAS in this case study, please see Appendices C.1.1 and C.1.4.

We chose the parameters for each of the four variants of CBS (value of the conditioning variable, if it is binary, and direction of effect) in order to search for systematic biases in COMPAS predictions and recommendations which disadvantage the protected class. For the separation scans, we detect positive deviations for the protected class attribute in the $\mathbb{E}(P \mid Y = 0, X)$ and $\Pr(P_{bin} = 1 \mid Y = 0, X)$, i.e., increase in predicted risk and increase in FPR for non-reoffending defendants, respectively. For the sufficiency scans, we detect a negative deviation for the protected class in the $\Pr(Y = 1 \mid P, X)$ and $\Pr(Y = 1 \mid P_{bin} = 1, X)$, i.e., decreased probability of reoffending conditional on predicted risk and on being flagged as high-risk, respectively. For all scans, we use all attributes except for the sensitive attribute when calculating the probability of being a member of the protected class (for the propensity score weighting step) and when generating the predicted values $\hat{I}$ in Section 3.2. All scans were run for 500 iterations with a penalty equal to 1.

Figure 3 contains the detected subgroups $S^*$, and their associated log-likelihood ratio scores $F(S^*)$ and corresponding indicators of statistical significance, found by each of the four variants of CBS, for various choices of the protected class: Black, white, female, male, younger (under the age of 25) and older (age 25+) defendants. Please see Appendix A.3 for the permutation test procedure used to determine statistical significance of CBS's detected biases. For the full set of results for all CBS scans when treating each attribute value as the protected class, please see Table 4 in Appendix C.1.2. This table includes information about the number of individuals and the observed rate (e.g., proportion of reoffending), both for the detected subgroup of the protected class, and for the corresponding

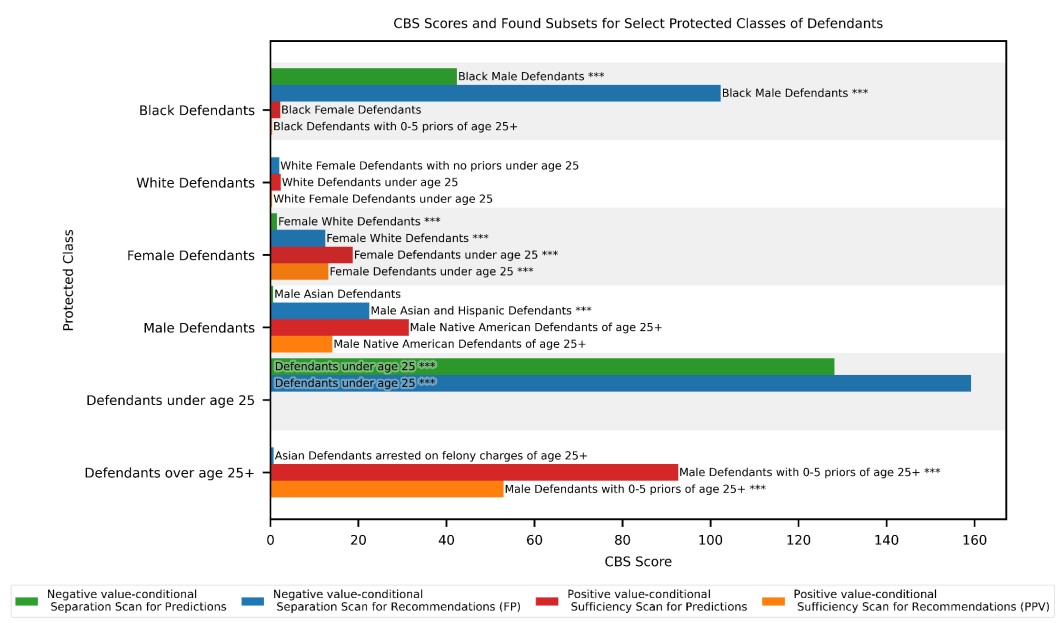

Figure 3: Scores of the subgroups found when running four variants of CBS on COMPAS data for different choices of protected class. A text description of the subgroup $S^*$ found for each scan is provided if the subgroup score $F(S^*)$ is greater than 0. *** indicates the subgroup's score is statistically significant with p-value < .05 measured by permutation testing, as described in Appendix A.3. We exclude statistically significant detected subgroups affected by over-estimation bias pertaining to Asian and Hispanic defendants because the $F(S^*)$ scores were small and visually challenging to display. Please reference Table 4 in Appendix C.1.2 for these results.

(comparison) subgroup of the non-protected class. For a discussion of the benchmark methodologies' results for COMPAS, please reference Appendix C.1.5. Below are the statistically significant racial and age biases that CBS found in COMPAS predictions and recommendations:

**Racial bias in COMPAS.** Figure 3 shows that the separation scans identify highly significant biases against a subgroup of Black defendants, while the sufficiency scans do not. These results support and complement the previous findings by ProPublica (Angwin et al., 2016b) and follow-up analyses (Chouldechova, 2017), which concluded that COMPAS has large error rate disparities which negatively impact Black defendants (corresponding to large scores for separation scans), and that its predictions are well-calibrated for Black defendants (corresponding to small scores for sufficiency scans). However, CBS's detected subgroup for the two separation scans adds a useful finding to this discussion: the large FPR disparity of COMPAS against Black defendants is even more significant in the intersectional subgroup of Black males. Non-reoffending Black male defendants have an FPR of 0.44, compared to non-reoffending non-Black male defendants' FPR of 0.19, whereas non-reoffending Black defendants have an FPR of 0.42, compared to non-reoffending non-Black defendants' FPR of 0.20. Sufficiency scans find Asian defendants arrested on misdemeanor charges have a lower rate of reoffending compared to non-Asian defendants with comparable COMPAS risk scores and Hispanic defendants flagged as high-risk by COMPAS have lower rate of reoffending compared to non-Hispanic defendants flagged as high-risk.

**Age bias in COMPAS.** Previous research argues that COMPAS relies heavily on the assumption that younger defendants are more likely to reoffend (Rudin et al., 2020), when computing risk scores. Younger defendants have a higher reoffending rate compared to older defendants (0.56 vs. 0.46), and thus, well-calibrated predictions and recommendations would result in younger defendants having higher FPR than older defendants. Our separation scans identify non-reoffending defendants under age 25 as the subgroup with the largest FPR disparity. On the other hand, our sufficiency scans identify a large subgroup bias within the protected class of defendants age 25+: older male defendants

with 0 to 5 priors have a lower rate of reoffending, as compared to younger male defendants with 0 to 5 priors, both for flagged high-risk defendants (sufficiency scan for recommendations) and for defendants with similar risk scores (sufficiency scan for predictions). This finding highlights the scenario described in Section 1 that CBS is designed to detect: predictions are well-calibrated between older and younger defendants, in aggregate, but not for the detected subgroup of older males with 0 to 5 priors.

For **gender bias in COMPAS**, reference Appendix C.1.3. For our **German Credit Data** case study, see Appendix C.2.

## 6 LIMITATIONS

Our CBS framework is designed to audit a classifier's predictions and recommendations for biases with respect to subgroups of a protected class, whereas competing methods provide mechanisms for both auditing and correcting classifiers. Combining auditors with correction and training presents the challenge of how to quantify the inherent trade-offs between performance and fairness when correcting for subgroup biases. Additionally, designing auditors that are linked to correction and training methods reinforces the framing that the primary solution to subgroup biases is to correct the models. Given that fairness is often context-specific, ideas of fairness could differ between stakeholders, and upstream biases exist in data sources used in many socio-technical settings, designing an optimally fair model is not always feasible. We endorse exploring larger policy shifts (not limited to model correction) to address biases that auditing tools like CBS might unearth that are correlated with broader societal issues.

CBS is designed to detect biases in the form of group fairness violations represented as conditional independence relationships. While CBS is easily generalizable to other objectives that can be represented as group-level conditional independence relationships, it is less generalizable to other fairness definitions such as individual and counterfactual fairness (Dwork et al., 2012; Kusner et al., 2017). Our technique for estimating the expectations $\hat{I}$ under the null hypothesis of no bias has the limitation (which is commonly cited in the average treatment effects literature) of only being reliable when using well-specified models for estimating the propensity scores of protected class membership and for estimating $\hat{I}$. Given the consistency of our COMPAS results in Section 5 with other researchers' findings about COMPAS, the process of estimating $\hat{I}$ seems to model the COMPAS data well. With that said, we encourage users of CBS to check estimates of $\hat{I}$ and if necessary, employ procedures common in the econometric literature (Imbens, 2004; Schuler and Rose, 2017) or calibration methods within the computer science literature. Lastly, there are various limitations to permutation testing, some of which are discussed in Berger (2000). For CBS specifically, if $\hat{I}$ is poorly estimated during permutation testing, this could result in higher type II errors where CBS is more likely to erroneously fail to reject the null hypothesis $H_0$ of no bias.

Our simulations in Section 4 account for bias in the form of shifts in the predicted and true probabilities (separately and jointly) – which produces predictive and aggregation biases – for a prescribed set of covariate attribute values in the protected class. We provide additional simulations with signal and base rate shifts represented as shifts in the true and predicted log-odds in Appendix B.4. In real-world scenarios, the generative process of bias might differ from the assumptions made in our simulations. Future research could determine and (if necessary) improve CBS's robustness to different generative schemas of bias. While this is a limitation of our simulations, the results of CBS for COMPAS, which is a real-world application where the biases present are not a result of our generative process, are in line with other research about biases in COMPAS and the U.S. criminal justice system (Chouldechova and G'Sell, 2017; Everett et al., 2011; Rudin et al., 2020). Additionally, we provide a discussion of the benchmark methodologies' results for COMPAS in Appendix C.1.5 to highlight that CBS has various advantages as an auditor in this real-world application (not restricted by the assumptions used in Section 4) compared to the benchmark methodologies' auditor results.

In summary, CBS is a flexible framework that works with most group-level fairness definitions to detect intersectional and contextual biases within subgroups of the protected class while overcoming some of the issues that arise when only considering fairness violations in aggregate for a single protected attribute value. CBS can discover intersectional and contextual biases in COMPAS scores and German Credit Data, and outperforms similar methods that audit classifiers for subgroup fairness.

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

## A    METHODS APPENDICES

### A.1    DETAILS ABOUT THE METHOD FOR GENERATING $\hat{I}$ USED IN SECTION 3.2 AND ITS LIMITATIONS

The method presented in Section 3.2 describes how to estimate $\hat{I}_i$, the expectation of the event variable $I_i$ for each individual $i$ in the protected class, under the null hypothesis, $H_0$, of no bias (i.e., $I \perp A \mid C, X$). Using the estimated $\hat{I}$ and observed $I$, we can determine which subgroups in the protected class have the largest deviations in $I$ as compared to what we would expect if there was no bias, $\hat{I}$. The method to generate $\hat{I}$ borrows from the literature on causal inference in observational settings, where propensity score reweighting is used to account for the selection of individuals into a "treatment" condition (here, membership in the protected class) given their observed covariates $X$.

The method to estimate $\hat{I}$ consists of the following steps:

1. Train a predictive model using all the individuals in the data to estimate $\Pr(A = 1 \mid X)$.

2. Use this model to produce the probabilities, $p_i^A = \Pr(A_i = 1 \mid X_i)$, and the corresponding propensity score weights, $w_i^A = \frac{p_i^A}{1 - p_i^A}$, for each individual $i$ in the non-protected class ($A_i = 0$). Intuitively, individuals in the non-protected class whose attributes $X_i$ are more similar to individuals in the protected class have higher weights $w_i^A$. This weighting scheme is used in the literature to produce causal effect estimates that can be interpreted as the average treatment effect on treated individuals (ATT) under typical assumptions of positivity and strong ignorability.

3. If the event variable, $I$, is binary (i.e., for all sufficiency scans and separation scan for recommendations), we train a model using only data for individuals in the non-protected class ($A_i = 0$) to estimate $\mathbb{E}_{H_0}[I \mid C, X]$ by weighting each individual $i$ in the non-protected class by $w_i^A$. The trained model is used to estimate the expectations $\hat{I}_i = \mathbb{E}_{H_0}[I_i \mid C_i, X_i]$ for each individual in the protected class ($A_i = 1$) under the null hypothesis, $H_0$, of $I \perp A \mid (C, X)$.

4. For the separation scan for predictions, we have a real-valued event variable, the probabilistic predictions $P$, rather than a binary event variable. We use a similar but modified process to estimate $\mathbb{E}_{H_0}[I \mid C, X]$, where $I = P$ and $C = Y$. For each individual $i$ in the non-protected class, we create two training records containing the same covariates $X_i$, but different labels and associated weights:

   (a) For the first record, we set the label, $I_{i_+}^{temp}$, equal to 1, and set the weight to $w_i^A P_i$.

   (b) For the second record, we set the label, $I_{i_-}^{temp}$, equal to 0, and set the weight to $w_i^A(1 - P_i)$

   We create a dataset that includes both records for each individual in the non-protected class and their associated weights, and use this concatenated data set to train a model that estimates $\mathbb{E}_{H_0}[I^{temp} \mid C, X]$, by weighting each individual $i$ in the non-protected class by either $w_i^A P_i$ or $w_i^A(1 - P_i)$ as described above. This approach is consistent with other CBS variants and enforces the desired constraint $0 \leq \hat{I}_i \leq 1$, unlike alternative approaches such as using regression models to predict $P$.

For value-conditional scans, CBS audits for biases in the subset of data where $C = z$, for $z \in \{0, 1\}$. Dataset $D$ is filtered before Step 3 to only include individuals where $C = z$. For example, for the value-conditional scan for FPR, we filter the data to only include individuals where $C = 0$ (or equivalently, $Y = 0$).

A probabilistic model can be used to estimate $\Pr(A = 1 \mid X)$ in Step 1, and a probabilistic model that allows for weighting of instances during training can be used to estimate $\mathbb{E}_{H_0}[I \mid C, X]$ in Steps 3 and 4. For Sections 4 and 5, as well as Appendices B.3 and B.4, we use logistic regression to

estimate $\Pr(A = 1 \mid X)$ and weighted logistic regression to estimate $\mathbb{E}_{H_0}[I \mid C, X]$. When estimating $\mathbb{E}_{H_0}[Y \mid P, X]$ (the realized expectation of $\mathbb{E}_{H_0}[I \mid C, X]$) for sufficiency scan for predictions, we transform the conditional variable, $P_i$, to its corresponding log-odds, $\log \frac{P_i}{1-P_i}$, prior to training, since we expect $\log \frac{Y_i}{1-Y_i}$ (the target of the logistic regression) to be approximately $\log \frac{P_i}{1-P_i}$ for well-calibrated classifiers. Alternative prediction models, such as random forests with Platt scaling for calibration of probability estimates, could also be used in place of logistic regression.

The method described above has the limitation of only producing accurate estimates of $\hat{I}$ when both the model for $\Pr(A = 1 \mid X)$ and $\mathbb{E}_{H_0}[I \mid C, X]$ are well-specified. Accurate estimates of $\hat{I}$ are essential for CBS to accurately detect the subgroup in the protected class with the most deviation between the observed $I$ and estimated $\hat{I}$ under the null hypothesis of no bias. Given the consistency of our findings for the COMPAS case study in Section 5 with other researchers' findings about COMPAS, as well as other checks we have performed to examine $\hat{I}$, we believe the method above suffices for COMPAS. However, we find that logistic regression does not do a good job of estimating $\hat{I}$ for the German Credit Data, due to the smaller dataset size and highly-correlated predictors. Thus we use a more flexible model—a gradient boosting classifier with Platt scaling—in our German Credit Data experiments in Appendix C.2 to ensure that CBS predictions are well-calibrated when computing propensity scores and when estimating $\hat{I}$. We encourage others using CBS to be aware of this limitation, pay special consideration to estimates of $\hat{I}$, and if necessary, employ methods from the causal inference literature on doubly robust estimation (Imbens, 2004; Schuler and Rose, 2017) or methods from the computer science literature for model calibration when producing estimates of $\hat{I}$.

Critically, we note that both discrete-valued and continuous-valued covariates $X_i$ can be used for estimating $\hat{I}$. Both the propensity model $\Pr(A = 1 \mid X)$ and the model of $\mathbb{E}_{H_0}[I \mid C, X]$ can incorporate either discrete-valued or continuous-valued covariates. However, continuous-valued covariates must be discretized or removed prior to the scan step, which assumes that all scan dimensions are discrete.

### A.2 FAST SUBSET SCANNING FOR CONDITIONAL BIAS SCAN

In this section, we explain the fast subset scanning (FSS) algorithm that CBS uses to find the subgroup of the protected class with the most biased predictions or recommendations (Neill, 2012). We will introduce FSS using a simplified example, for illustrative purposes, to highlight the computational difficulties inherent in subset scanning, the additive property of the score functions for CBS that enable computationally feasible subset scanning, and the implementation of FSS for CBS.

Let us assume a dataset of individuals in the protected class ($A = 1$), denoted as $Q = \{(X^1, I, \hat{I})\}$, that contains values of the event variable $I_i$, estimates $\hat{I}_i$ of the expected value of the event variable under the null hypothesis of no bias, and a single categorical covariate attribute $X_i^1$ for each individual $i$. For concreteness, we perform a sufficiency scan for predictions, therefore, the event variable $I_i$ is the observed binary outcome $Y_i$ for individual $i$, and the corresponding $\hat{I}_i$ is the estimated $\Pr(Y_i = 1 \mid P_i, X_i)$ under the null hypothesis $H_0$ that $Y \perp A \mid (P, X)$. $S$ refers to a subgroup of $Q$, which in our simple example is a non-empty subset of values for attribute $X^1$. Since our event variable is binary, we use the Bernoulli likelihood function to represent the hypotheses in the score function, $F(S)$, used to determine the level of anomalousness of a subgroup $S$ of $Q$.

In the worst-case scenario, $X^1$ could be a categorical variable with distinct values for each of the $n$ rows of data in $Q$. If we were to score all of the possible $S \subseteq Q$ using $F(S)$, this method would have a runtime of $O(2^n)$, which would be computationally infeasible. To overcome this computational barrier, FSS relies on its score functions, $F(S)$, being a part of an efficiently optimizable class of functions in order to find the most anomalous subset $S^* = \arg\max_{S \subseteq Q} F(S)$ without the need to evaluate all of the subsets of $Q$. The property that determines if a function is a part of this class that enables fast subset scanning is called Additive Linear-Time Subset Scanning (ALTSS) (Speakman et al., 2016) and is formally defined below. Informally, if $F(S)$ can be represented as an additive set function over all instances $i \in S$ when conditioning on the free parameter ($q$ for the Bernoulli distribution or $\mu$ for the Gaussian distribution in Table 2), it satisfies this property (Speakman et al., 2016).

To explore how FSS exploits the ALTSS property for computationally efficient subset scanning, assume that the categorical covariate $X^1$ for each individual $i$ can only be equal to one of four values,

$X_i^1 \in \{a, b, c, d\}$. FSS constructs a subset for each attribute value of $X^1$ such that $S_a = \{i \in Q : X_i^1 = a\}$, $S_b = \{i \in Q : X_i^1 = b\}$, $S_c = \{i \in Q : X_i^1 = c\}$, $S_d = \{i \in Q : X_i^1 = d\}$. Since we are using the likelihood function for the Bernoulli distribution for $F(S)$, $F(S)$ is a concave function of the free parameter $q$, and for illustrative purposes, we will assume that $\max_q F(S)$ is positive for all subsets $S_a, S_b, S_c$ and $S_d$. Therefore, for each subset $S_a, S_b, S_c$ and $S_d$, $F(S)$ is a function over the domain of $q$, where as $q$ increases from $-\infty$, $F(S)$ eventually equals 0 and then the global maximum for $F(S)$ for that given subset, and then starts decreasing until it again reaches a point where $F(S) = 0$, and then remains negative as $q$ approaches $\infty$. FSS identifies three $q$ values for each subset, $S \in \{S_a, S_b, S_c, S_d\}$:

1. The first value of $q$ where $F(S) = 0$ as $q$ increases from $-\infty$ to $\infty$, which we will refer to as $q_{min}$.

2. The second value of $q$ where $F(S) = 0$ as $q$ increases from $-\infty$ to $\infty$, which we will refer to as $q_{max}$.

3. The value of $q$ for $\arg\max_q F(S)$, which we will refer to as $q_{\text{MLE}}$.

Each distinct $q_{min}$ and $q_{max}$ value for subsets $(S_a, S_b, S_c, S_d)$ is a value of $q$ where the score function $F(S)$ becomes negative or positive for at least one of these four subsets. By sorting all of the distinct $q_{min}$ and $q_{max}$ values across all the subsets $(S_a, S_b, S_c, S_d)$ in ascending order, we construct a list of $q$ values, $\{q_{(1)}, ..., q_{(m)}\}$, where each pair of adjacent values, $q_{(k)}$ and $q_{(k+1)}$, represents an interval of the $q$ domain, $(q_{(k)}, q_{(k+1)})$, for which each subset $S \in \{S_a, S_b, S_c, S_d\}$ has either $F(S) > 0$ for the entire interval or $F(S) < 0$ for the entire interval. For each interval, we perform the following:

1. Find the midpoint of the interval (average of $q_{(k)}$ and $q_{(k+1)}$), which we refer to as $q_k^{\text{mid}}$.

2. Create a new subset $S_k^{\text{aggregate}}$ by aggregating all subsets $S \in \{S_a, S_b, S_c, S_d\}$ where the subset's $q_{min} < q_k^{\text{mid}}$ and the subset's $q_{max} > q_k^{\text{mid}}$, i.e., $F(S) > 0$ when $q = q_k^{\text{mid}}$ and therefore for the entire interval $(q_{(k)}, q_{(k+1)})$. Since the score function is additive, conditioned on $q$, we know that a subset $S$ will make a positive contribution to the score $F(S_k^{\text{aggregate}})$ if and only if $F(S) > 0$ for that value of $q$. Thus, we know that the highest scoring subset $S_k^{\text{aggregate}}$ for that interval $[q_{(k)}, q_{(k+1)}]$ contains all and only those subsets $S$ with $F(S) > 0$ at $q = q_k^{\text{mid}}$.

3. Find the maximum likelihood estimate of $q$, $q_{\text{MLE}}^{\text{aggregate}} = \arg\max_q F(S_k^{\text{aggregate}})$, and the corresponding score $F(S_k^{\text{aggregate}})$.

The aggregate subset, $S_k^{\text{aggregate}}$, with the highest score for $F(S)$ using its associated $q_{\text{MLE}}^{\text{aggregate}}$ is the most anomalous subset when considering subsets formed by combinations of different attribute-values of $X^1$.

For our simplified example, there are at most 8 distinct $q_{min}$ or $q_{max}$ values from the four subsets $(S_a, S_b, S_c, S_d)$, and thus at most 7 distinct intervals $(q_{(k)}, q_{(k+1)})$ that must be considered. For a given interval, we need to evaluate only a single subset $S_k^{\text{aggregate}}$, and thus, only 7 of the 15 non-empty subsets of $\{S_a, S_b, S_c, S_d\}$. More generally, if $n$ is the arity (number of attribute values) of categorical attribute $X^1$, at most $2n - 1$ of the $2^n - 1$ non-empty subsets of attribute values must be evaluated to identify the highest-scoring subgroup.

The scenario where the covariates consist of a single categorical attribute is a simplified example, where only a single iteration of FSS is needed to find the optimal subset, $S^*$, of $Q$. When there are two or more attributes for the covariates, multiple iterations of FSS must be performed to find the optimal subset. On each iteration the following is performed:

1. We define an initial subset, $S_{temp}$ where:

   (a) If it is the first iteration, all of the attribute values for each attribute are included in $S_{temp}$.

   (b) Otherwise, a random subset of attribute values for each attribute are chosen to be included in $S_{temp}$.

2. For each attribute $X^i$, in random order, we construct subsets by partitioning $S_{temp}$ by the distinct attribute values of $X^i$, form intervals across the domain of $q$ for $F(S)$, and then

assemble and score the subsets for each interval (as described above). $S_{temp}$ is updated as higher scoring subsets using $F(S)$ are found. Therefore, when an attribute is evaluated, $S_{temp}$ contains only rows of $Q$ that fit the found criteria (in the form of attribute values) from previously evaluated attributes, excluding the attribute currently under consideration. This iterative ascent procedure is repeated until convergence.

Multiple iterations are performed with the final optimal subset being the subset with the highest score using $F(S)$ found across all the iterations, $S^*$. For the pseudocode of FSS for CBS, please see Algorithm 1. The final results from FSS are the optimal subset, $S^*$, in the form of attribute-values that form the criteria for the subgroup in the protected class with the most anomalous bias detected, the parameter $q$ or $\mu$ that maximizes $F(S^*)$, and the score $F(S^*)$ given the parameter $q$ or $\mu$.

### A.2.1 FORMAL DEFINITION OF ADDITIVE LINEAR-TIME SUBSET SCANNING PROPERTY (ALTSS)

Below we provide a formal definition of the Additive Linear-Time Subset Scanning Property. The score functions, $F(S)$, used to evaluate subgroups are a log-likelihood ratio formed from two different hypotheses whose likelihoods are modeled by likelihood functions for either the Bernoulli distribution or Gaussian distribution, both of which satisfy the Additive Linear-time Subset Scanning Property (Speakman et al., 2016; Zhang and Neill, 2016).

**Definition A.1** (Additive Linear-time Subset Scanning Property). A function, $F : S \times \theta \to \mathbb{R}_{>0}$, that produces a score for a subset $S \subseteq D$, where $D$ is a set of data and $\theta = \arg\max_\theta F(S \mid \theta)$, satisfies the Additive Linear-time Subset Scanning Property if $F(S \mid \theta) = \sum_{s_i \in S} F(s_i \mid \theta)$ where $s_i$ is a subset of $S$ and $\forall s_i, s_j \in S$, where $s_i \neq s_j$, we have $s_i \cap s_j = \emptyset$.

We refer to the score functions, $F(S)$, contained in the rightmost column of Table 2 as $F(S \mid \mu)$ for the score functions that use the Gaussian likelihood function to form hypotheses and $F(S \mid q)$ for the score functions that use the Bernoulli likelihood function to form hypotheses. $F(S \mid q)$ contains a summation, $\sum_{i \in S}(I_i \log q - \log(q\hat{I}_i - \hat{I}_i + 1))$, that is the sum of individual-specific values derived from $I_i$, $\hat{I}_i$, and $q$. Given that each individual is distinct, $F(S \mid q) = \sum_{i \in S} F(s_i \mid q)$, where $s_i$ is the subset of $S$ that contains only individual $i$, satisfies the ALTSS property. Similarly, $F(S \mid \mu)$ contains a summation, $\sum_{i \in S} \Delta_i$, that is the sum of individual-specific values $\Delta_i$ derived from $I_i$, $\hat{I}_i$, and $\mu$. Therefore $F(S \mid \mu) = \sum_{s_i \in S} F(s_i \mid \mu)$, where $s_i$ is the subset of $S$ that contains only individual $i$, satisfies the ALTSS property.

### A.2.2 PSEUDOCODE OF FAST SUBSET SCAN ALGORITHM FOR CONDITIONAL BIAS SCAN

Algorithm 1 is the pseudocode for the Fast Subset Scan (FSS) algorithm used in the CBS framework (Neill, 2012). The algorithm finds the subgroup, $S^*$, with the most anomalous signal (i.e., the highest score $F(S^*)$) in a dataset. For CBS, this signal is in the form of a bias (according to one of the fairness definitions in Table 1) against members of the protected class ($A = 1$) for subgroup $S^*$. The dataset passed to the FSS algorithm by CBS contains only individuals $i$ in the protected class, and FSS compares their values of the event variable $I_i$ to the estimated expectations $\hat{I}_i$ under the null hypothesis of no bias.

At the initialization of FSS, placeholder variables are created that will hold the most anomalous subset ($S^*$), and the subset's corresponding information ($\theta^*$, $Score^*$), across all iterations (Lines 1-3). At the beginning of an iteration, a random subset is picked (set of attribute-values) as the starting subset, $S_{temp}$, with the exception of the first iteration where the starting subset includes all attribute values, as shown in the if-else statement starting on Line 5. For each iteration of this algorithm, we repeatedly choose a random attribute to scan (i.e., we scan over subsets of its attribute values) as shown in Lines 14-15, until convergence (i.e., when all attributes have been scanned without increasing the score $F(S_{temp})$).

For each attribute $X_{temp}$ to be scanned, for each of its attribute values $X_{temp_i}$, we score the subset $S_{X_{temp_i}}$ containing only the records with the given value of that attribute ($X_{temp} = X_{temp_i}$), and matching subset $S_{temp}$ on all other attributes in $X$. We write this as $S_{X_{temp_i}} \leftarrow S_{temp}^{relaxed} \cap \{i \in D : X_{temp} = X_{temp_i}\}$, where $S_{temp}^{relaxed}$ is the relaxation of subset $S_{temp}$ to include all values for attribute $X_{temp}$. Along with scoring this attribute-value subset $S_{X_{temp_i}}$, we find the two values of $\theta$

---

**Algorithm 1** Fast Subset Scan for Conditional Bias Scan

---

**Require:** $n_{iters} > 0, (X_i, \hat{I}_i, I_i) \, \forall i \in D$ where $A_i = 1, direction \in \{\text{positive}, \text{negative}\}$
1: $S^* \leftarrow \{\}$
2: $Score^* \leftarrow -\infty$
3: $\theta^* \leftarrow -\infty$
4: **for** $j \leftarrow 1 \ldots n_{iters}$ **do**
5:     **if** $j == 1$ **then**
6:         $S_{temp} \leftarrow$ all attribute-values for each attribute in $X$
7:     **else**
8:         $S_{temp} \leftarrow$ random nonempty subset of attribute-values for each attribute in $X$
9:     **end if**
10:     $\theta_{temp} \leftarrow \arg\max_\theta(F(S_{temp} \mid \theta))$
11:     $Score_{temp} \leftarrow F(S \mid \theta_{temp})$
12:     $n_{attributes} \leftarrow$ number of attributes in $X$
13:     $n_{scanned} \leftarrow 0$                                        ▷ mark all attributes as unscanned
14:     **while** $n_{scanned} < n_{attributes}$ **do**
15:         $X_{temp} \leftarrow$ randomly selected attribute that is marked as unscanned
16:         **for** $X_{temp_i} \in X_{temp}$ **do**                       ▷ for all attribute-values in $X_{temp}$
17:             $S_{X_{temp_i}} \leftarrow S_{temp}^{relaxed} \cap \{i \in D : X_{temp} = X_{temp_i}\}$    ▷ see Appendix A.2.2 for definition of $S_{temp}^{relaxed}$
18:             $\theta_{min_i}, \theta_{max_i} \leftarrow \arg_\theta(F(S_{X_{temp_i}} \mid \theta) = 0)$    ▷ exception noted in Appendix A.2.2
19:             $\theta_{MLE_i} = \arg\max_\theta(F(S_{X_{temp_i}} \mid \theta))$
20:             $Score_i \leftarrow F(S_{temp_i} \mid \theta_{MLE_i})$
21:             Adjust $\theta_{min_i}$ and $\theta_{max_i}$ depending on the $direction$ of scan  ▷ explained in text of Appendix A.2.2
22:         **end for**
23:         $\theta_{intervals} \leftarrow \{\theta_{min_i}, \theta_{max_i} \forall X_{temp_i} \in X_{temp}\}$ in ascending order       ▷ all values of $\theta$ where $F(S) = 0 \, \forall X_{temp_i} \in X_{temp}$, indexed by $\theta_{(k)}$ below
24:         $Score_{interval} \leftarrow -\infty$
25:         $S_{interval} \leftarrow \{\}$
26:         $\theta_{interval} \leftarrow -\infty$                                   ▷ not to be confused with $\theta_{intervals}$
27:         **for** $k \leftarrow 1 \ldots length(\theta_{intervals}) - 1$ **do**
28:             $S_k^{\text{aggregate}} \leftarrow \{\}$
29:             $\theta_k^{\text{mid}} \leftarrow \frac{\theta_{(k)} + \theta_{(k+1)}}{2}$
30:             **for** $X_{temp_i} \in X_{temp}$ **do**
31:                 **if** $Score_i > 0$ and $\theta_{min_i} < \theta_k^{\text{mid}}$ and $\theta_{max_i} > \theta_k^{\text{mid}}$ **then**
32:                     $S_k^{\text{aggregate}} \leftarrow S_k^{\text{aggregate}} \cup S_{X_{temp_i}}$
33:                 **end if**
34:             **end for**
35:             $\theta_k^{\text{aggregate}} \leftarrow \arg\max_\theta(F(S_k^{\text{aggregate}} \mid \theta))$
36:             $Score_k^{\text{aggregate}} \leftarrow F(S_k^{\text{aggregate}} \mid \theta_k^{\text{aggregate}})$
37:             **if** $Score_k^{\text{aggregate}} > Score_{interval}$ **then**
38:                 $Score_{interval} \leftarrow Score_k^{\text{aggregate}}$
39:                 $S_{interval} \leftarrow S_k^{\text{aggregate}}$
40:                 $\theta_{interval} \leftarrow \theta_k^{\text{aggregate}}$
41:             **end if**
42:         **end for**

---

```
43:          if Score_temp < Score_interval then
44:              Score_temp ← Score_interval
45:              S_temp ← S_interval
46:              θ_temp ← θ_interval
47:              n_scanned ← 0                                    ▷ mark all attributes as unscanned
48:          end if
49:          n_scanned ← n_scanned + 1                            ▷ mark attribute X_temp as scanned
50:      end while
51:      if Score* < Score_temp then
52:          Score* ← Score_temp
53:          S* ← S_temp
54:          θ* ← θ_temp
55:      end if
56: end for
57: return S*, Score*, θ*
```

where $F(S_{X_{temp_i}}) = 0$, $\theta_{min_i}$ and $\theta_{max_i}$, and the $\theta$ that maximizes $F(S_{X_{temp_i}})$, $\theta_{MLE_i}$, with the exception of attribute-value subsets $S_{X_{temp_i}}$ that are not positive for any value of $\theta$. This is shown in the for-loop in Lines 16-21.

Line 21 states that $\theta_{min_i}$ and $\theta_{max_i}$ must be adjusted according to the direction of the scan to enforce that the found parameters $\theta_{min_i}$ and $\theta_{max_i}$ adhere to the restrictions set by the direction of the scan. The constraints necessary for the scans to detect biases in the positive and negative directions are fully specified in Table 2. For positive scans that have score functions that utilize the Gaussian likelihood function to form hypotheses, $\theta_{min_i} = \max(0, \theta_{min_i})$ and for negative scans that utilize the Gaussian likelihood function, $\theta_{max_i} = \min(0, \theta_{max_i})$. For positive scans that have score functions that utilize the Bernoulli likelihood function to form hypotheses, $\theta_{min_i} = \max(1, \theta_{min_i})$ and for negative scans that utilize the Bernoulli likelihood function, $\theta_{max_i} = \min(1, \theta_{max_i})$. Attribute-value subsets $S_{X_{temp_i}}$ should not be considered when choosing subsets for $S^{\text{aggregate}}$ for positive scans where $\theta_{max_i} < 0$ or $\theta_{max_i} < 1$ for scans using the Gaussian likelihood function or Bernoulli likelihood function in $F(S)$, respectively. Conversely, attribute-value subsets $S_{X_{temp_i}}$ should not be considered when choosing subsets for $S^{\text{aggregate}}$ for negative scans where $\theta_{min_i} > 0$ or $\theta_{min_i} > 1$ for scans using the Gaussian likelihood function or Bernoulli likelihood function in $F(S)$, respectively.

We sort the $\theta_{min_i}$ and $\theta_{max_i}$ values found across all the attribute values of the attribute we are scanning in ascending order in Line 23. These form a list of intervals over the domain of $\theta$. For each interval, we calculate a midpoint of that interval, and aggregate all the attribute-value subsets that have a positive score, $F(S)$, when $\theta$ equals the midpoint of that interval in Lines 30-33. If the aggregated subset of attribute values with the maximum score across all the intervals is greater than the score of $S_{temp}$, we update $S_{temp}$ and all of its accompanying information ($\theta_{temp}$, $Score_{temp}$) to equal the maximum-scoring subset of aggregated attribute-values across all the intervals and its accompanying information. $S_{temp}$ is continuously updated as higher scoring subsets are found as we scan over all the attributes and their attribute values.

At the end of an iteration, if the found subset, $S_{temp}$, has a higher score than the global maximum scoring subset $S^*$, then $S^*$ and its accompanying information ($\theta^*$, $Score^*$) are replaced with $S_{temp}$ and $S_{temp}$'s accompanying information. Once all the iterations have completed, the subset with the maximum score found across all iterations is returned, $S^*$, with its score $F(S^*|\theta^*)$ and accompanying $\theta^*$ parameter.

McFowland III et al. (2023) show that a similar multidimensional scan algorithm, used for heterogeneous treatment effect estimation, will converge with high probability to a near-optimal subset when run with multiple iterations.

### A.3 Permutation Testing to Determine Statistical Significance of Detected Subgroups

As discussed in Section 3.3, the statistical significance ($p$-value) of the discovered subgroup $S^*$ can be obtained by *permutation testing*, which correctly adjusts for the multiple testing resulting from

searching over subgroups. To do so, we generate a large number of simulated datasets under the null hypothesis $H_0$, perform the same CBS scan for each null dataset (maximizing the log-likelihood ratio score over subgroups, exactly as performed for the original dataset), and compare the maximum score $F(S^*)$ for the true dataset to the distribution of maximum scores $F(S^*)$ for the simulated datasets. The detected subgroup is significant at level $\alpha$ if its score exceeds the $1 - \alpha$ quantile of the $F(S^*)$ values for the simulated datasets. To generate each simulated dataset, we copy the original dataset and randomly permute the values of $A_i$ (whether or not each individual is a member of the protected class), thus testing the null hypothesis that $A$ is conditionally independent of the event variable $I$.

This permutation testing approach is computationally expensive, multiplying the runtime by the total number of datasets (original and simulated) on which the CBS scan is performed, but it has the benefit of bounding the overall false positive rate (family-wise type I error rate) of the scan while maintaining high detection power. In comparison, a simpler approach like Bonferroni correction would also bound the overall false positive rate, and would require much less runtime, but would suffer from dramatically reduced detection power. For a given dataset, the score threshold for significance at a fixed level $\alpha = .05$ will differ for different choices of the sensitive attribute and protected class. Thus, if CBS is used to audit a classifier for possible biases against multiple protected classes, a separate permutation test must be performed for each protected class value.

### A.4    Conditional Bias Scan Framework Parameters

Table 3 contains all the parameters needed to run Conditional Bias Scan.

## B    Evaluation Appendices

### B.1    Adaptations of the Benchmark Methods used in Evaluation

Both GerryFair and MultiAccuracy Boost provide implementations of their methods on GitHub (Neel et al., 2019; Kim et al., 2019b). Our goal was to use their provided code with minimal changes as benchmarks in Sections 4 and 5. However, GerryFair and MultiAccuracy Boost do not provide the functionality to indicate whether to audit for bias in the positive direction (under-estimation bias) or bias in the negative direction (over-estimation bias). This lack of functionality makes results from CBS substantially different than those returned by GerryFair and MultiAccuracy Boost.

For GerryFair's auditor, given the type of error rate to audit (false negative rate or false positive rate), they train four linear regressions using the features ($X$) as dependent variables with the following four sets of labels:

1. Two linear regressions with the zero set as labels.

2. One linear regression with the labels set to a measurement that assigns positive costs for predictions that deviate in the *positive* direction (when the predictions are greater than the observed global error rate), and negative costs otherwise.

3. One linear regression with the labels set to a measurement that assigns positive costs for predictions that deviate in the *negative* direction (when the predictions are less than the observed global error rate), and negative costs otherwise.

They use the predictions from the linear regressions to flag a subset of data where the predictions from the linear regression trained with the zero set labels are greater than the values predicted by the linear regression trained with the costs representing deviations of the predictions from the observed baseline error rate metric of interest as labels. Two linear regressions are used to estimate deviations of the predictions from the observed error rate baseline, and therefore they form two subgroups: (1) a subgroup with rows that are estimated to have predictions that are greater than the baseline for the metric of interest; and (2) a subgroup with rows that are estimated to have predictions that are less than the baseline for the metric of interest. The original GerryFair implementation uses a heuristic to decide which subgroup has more significant biases and returns that subgroup accordingly. The subgroup with the rows that are estimated to have predictions that are greater than the metric of interest more closely aligns with the concept of auditing for bias in the positive direction or auditing for under-estimation bias. Since CBS provides the functionality of auditing for biases of a specific

Table 3: Table with all parameters needed to run Conditional Bias Scan.

| Parameter | Purpose | Parameter Attribute Values | Sections for Reference |
|---|---|---|---|
| Membership in Protected Class Indicator Variable ($A$) | Binary attribute which defines whether each individual is a member of the protected class. We wish to identify any biases that are present in the classifier's predictions or recommendations that impact the protected class. | | 3 |
| Scan Type | The subcategory of the scan type | Separation scan for recommendations; Separation scan for predictions; Sufficiency scan for recommendations; Sufficiency scan for predictions | 3.1 |
| Event Variable ($I$) | The event of interest for the scan. The abstracted event variable must be defined as either the outcome, prediction, or recommendation variable. | $Y$; $P$; $P_{bin}$ | 3, 3.1 |
| Conditional Variable ($C$) | The conditional variable for the scan. The abstracted conditional variable must be defined as either the outcome, prediction, or recommendation variable. | $Y$; $P$; $P_{bin}$ | 3, 3.1 |
| Field value ($z$) of Conditional Variable ($C = z$) | For value-conditional scans, this is the value on which we are conditioning the conditional variable ($C$). Defining a field value results in scans that detect different forms of fairness violations. | None; 0; 1 | 3, 3.2, 3.3, A.2 |
| List of Attributes for forming subgroups ($X$) | List of attributes to scan over to form subgroups | | 3, 3.1, A.2 |
| Direction of Bias | Specifying whether we are detecting under-estimation bias (positive direction) or over-estimation bias (negative direction) | Positive; Negative | 3.1, 3.3, A.2 |
| List of Attributes for estimating $\hat{I}$ ($X$) | List of attributes used for conditioning when producing $\hat{I}$. In this paper we use the same attributes to form subgroups and produce $\hat{I}$. This does not necessarily have to be the case for all applications of CBS. | | 3.2, A.1 |
| Subgroup Complexity Penalty | The non-negative integer-valued scalar penalty that is subtracted from the score function for each subgroup, depending on the subgroup's total number of included values for each covariate $X^1 \ldots X^m$, not including covariates for which all values are included. | 0+ (default value: 1) | 3.3 |
| Scan Iterations | Specifies the number of iterations to run the fast subset scanning algorithm | 1+ (default value: 500) | 3.3, A.2 |

The table lists the parameter, purpose of the parameter, possible values of the parameter, when applicable, and the sections in our paper where this parameter is described in further detail.

direction, we add an option to GerryFair that allows the user to determine which direction of bias they are interested in, making GerryFair's results more comparable to CBS.

For each simulation, we ran GerryFair two times, once to detect bias in the form of systematic increases in the false positive rate, and once to detect bias in the form of systematic increases in the false negative rate. In each case, we allow GerryFair to use all covariates ($X$) to make the predictions used to form subgroups, including the protected class category. This resulted in two result sets for GerryFair for each simulation. We present the result set in Section 4 that had the highest overall accuracy for most of the simulations, which is the GerryFair setup for detecting increased false positive rate. GerryFair returns a subgroup that could contain individuals in both the protected class and the non-protected class. To have the accuracy measurements for GerryFair and CBS be comparable, we filter the subgroup returned by GerryFair to only include individuals in the protected class before calculating the subgroup's accuracy.

MultiAccuracy Boost is an iterative algorithm where, on each iteration, it audits for a subgroup with inaccuracies and then corrects that subgroup's predicted log-odds. More specifically, for each iteration:

1. A custom heuristic is calculated for all rows of data, similar to an absolute residual, where larger values represent a larger deviation between the observed labels and predictions.

2. The residuals of all the rows' predictions and observed outcomes are calculated.

3. The full data is split into a training and holdout set.

4. Three partitions of data are created for the training data, hold out data, and the full dataset:

    (a) A partition containing all the rows.
    (b) A partition containing all the rows with predictions greater than 0.50.
    (c) A partition containing all the rows with predictions less than or equal to 0.50.

5. For each of the partitions of data constructed in Step 4:

    (a) A ridge regression classifier (using $\alpha = 1.0$) is trained using the respective partition in the training data, with the covariates $X$ and the sensitive attribute $A$ as features and the custom heuristic calculated in Step 1 as labels.
    (b) The ridge regression classifier is used to make predictions for the respective partition in the holdout data.
    (c) If the average of the predictions multiplied by the residuals for the partition set in the hold out data is greater than $10^{-4}$, then the predicted log-odds for the respective partition in the full dataset is shifted by the predictions multiplied by 0.1.
    (d) If the predicted log-odds are updated, the iteration terminates and no other partitions of data are evaluated for that iteration.

The steps above are slightly modified for the scenario of a classifier that produces a singular probability of a positive outcome whereas the original MultiAccuracy Boost was designed for was a bivariate outcome vector from a Inception-ResNet-v1 model. To make MultiAccuracy Boost audit for bias in one direction, when calculating whether a partition of the data's predicted log-odds should be updated using the holdout data to remove an inaccuracy, we override the residuals that are negative with 0. In effect, we only consider rows with negative outcomes when deciding which partition of predictions have inaccuracies that need to be corrected on a given iteration. This was the least invasive modification we could make to MultiAccuracy Boost to have it solely consider bias in the positive direction when deciding which subgroup's predicted log-odds to update. When using this slight adaptation, we see an increase in the overall average accuracy for the simulations by approximately 8% for MultiAccuracy Boost compared to a version of MultiAccuracy Boost without the modification intended to account for directional bias.

Since the auditor and correction method are functioning in tandem, we run all iterations of the algorithm and log each subgroup (i.e., partition) that was detected as needing a correction to its predicted log-odds and its associated score calculated in Step 5c. After the algorithm terminates, we find the partition with the highest score and return its associated partition in the full data set. The decision to return the partition with the highest score across all the iterations of MultiAccuracy Boost in the simulations is motivated by the fact that MultiAccuracy Boost's auditor has no theoretical guarantees of detecting the most inaccurate partition on a specific iteration of the algorithm. Similarly to GerryFair, MultiAccuracy Boost detects a subgroup that contains members of the protected class and non-protected class. We filter all the individuals in the returned subgroup to only contain individuals who are part of the protected class before calculating the accuracy of the returned partition.

One distinction between these methods and CBS is that their auditors were intended to be used in conjunction with another process to improve a classifier or predictions. Therefore, their auditors were designed to have the level of detection accuracy necessary to discern which subgroups or partitions of data need to be corrected, either by modifying the classifier or by post-processing their predicted log-odds. Given that both methods suggest that they can be used for auditing purposes, they are appropriate choices as benchmarks for CBS, but it is important to note that CBS was specifically designed to have a high accuracy for bias detection, whereas that was not necessarily an explicit intention of GerryFair or MultiAccuracy Boost.

### B.2 Explanation of the Additive Term ($\epsilon^{true}$) for the True Log-Odds used in the Generative Model for the Semi-synthetic Data

For the evaluation simulations described in Section 4, when producing the true log-odds that are used to determine the outcomes and predicted values, we add a term to each row's true log-odds of a value drawn from a Gaussian distribution $\epsilon_i^{true} \sim \mathcal{N}(0, \sigma_{true})$ where $\sigma_{true} = 0.6$. We add this term to the true log-odds to ensure that when the true probabilities (expit($L_i^{true}$)) for the rows of $S_{bias}$ in the protected class are injected with $\mu_{suf}$, this results in a violation of the fairness definition for sufficiency.

For the remainder of this section we will focus on sufficiency scan for predictions, but our explanation below is applicable for sufficiency scan for recommendations as well. Sufficiency implies that the outcomes $Y$ are conditionally independent of membership in the protected class $A$ given the predictions $P$ and covariates $X$, that is, $Y \perp A \mid (P, X)$. Assume that we have predictions that are independent of the outcome conditional on the covariates, $Y \perp P \mid X$. Since the outcome is independent of the predictions conditional on the covariates, the definition of sufficiency simplifies to $Y \perp A \mid X$. This simplification of sufficiency reduces sufficiency scans to finding the subgroup in the protected class with the largest base rate difference from its corresponding subgroup in the non-protected class regardless of that subgroup's predictions. Therefore, it is not evaluating sufficiency violations because these base rate differences are independent of the predictions. Consequentially, when there is *no* base rate difference between the protected and non-protected class conditional on the covariates, $(Y \perp A \mid X)$, in order for sufficiency to be violated, $Y \not\perp A \mid (P, X)$, we must also have $Y \not\perp P \mid X$. This is formally stated in Theorem B.1.

**Theorem B.1.** *To have violations of the sufficiency definition, $Y \not\perp A \mid (P, X)$, when there are no base rate differences between the protected class and non-protected class conditional on the covariates, $Y \perp A \mid X$, the predictions and outcomes must be conditionally dependent given the covariates, $Y \not\perp P \mid X$.*

*Proof.* Let us assume that (i) there are no base rate differences between protected and non-protected class conditional on the covariates, $Y \perp A \mid X$; (ii) outcomes are independent of the predictions conditional on the covariates, $Y \perp P \mid X$; and (iii) violations of the sufficiency definition exist, $Y \not\perp A \mid (P, X)$. We will show that these three statements lead to a contradiction. First, $(Y \perp P \mid X)$ and $(Y \perp A \mid X)$ together imply that $Y \perp (P, A) \mid X$. Furthermore, using the weak union axiom for conditional independence, $Y \perp (P, A) \mid X$ implies that $Y \perp A \mid (P, X)$, which contradicts (iii). Since these three statements cannot all be true, we know that no base rate differences (i) and violations of sufficiency (iii) together imply that the outcomes cannot be independent of the predictions conditional on the covariates, $Y \not\perp P \mid X$. $\square$

To ensure that $Y \not\perp P \mid X$, the predictions $P$ must carry information about the outcomes $Y$ that is not carried in $X$. By adding the term $\epsilon_i^{true}$ to the true log-odds for each row, given that the predicted log-odds (and the corresponding predicted probabilities $P_i$ and binarized recommendations $P_{i,bin}$) and the outcomes $Y$ are both derived from the true log-odds, this ensures that $Y \not\perp P \mid X$ in the evaluation simulations because $P$ carries information about $Y$, in the form of the added row-wise terms (drawn from a Gaussian distribution), that are captured in $Y$, but are not captured in $X$.

### B.3 Additional Evaluation Simulations

To evaluate (Q3) in Section 4, we modify the characteristics of $S_{bias}$, by varying $n_{bias}$ and $p_{bias}$ for three settings, when $\mu_{sep} = 0.50$, $\mu_{suf} = 0.50$, and $\delta = 0.25$. For each setting, we perform two

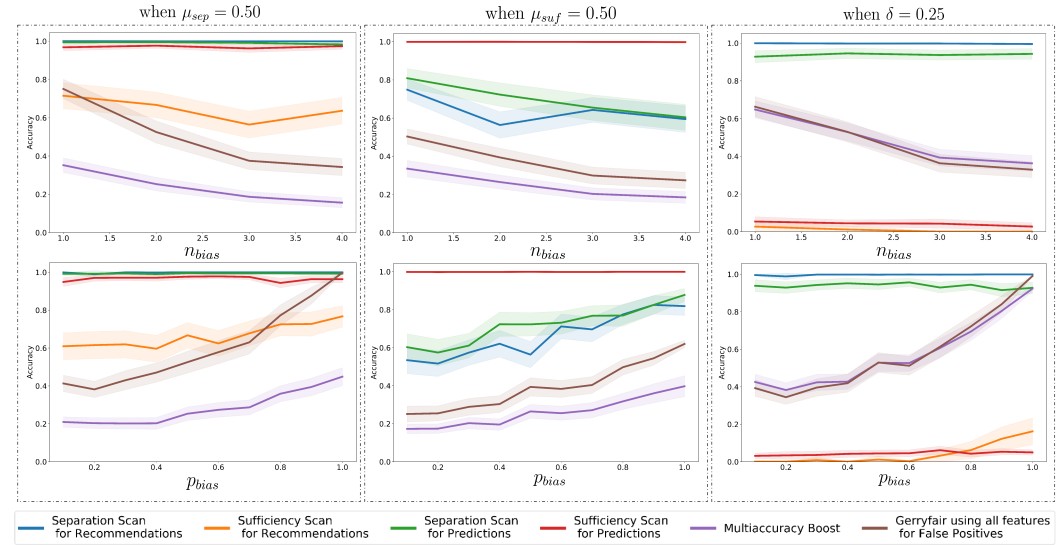

Figure 4: Average accuracy (with 95% CI) for biases and base rate shifts injected into subgroup $S_{bias}$ of the protected class, for CBS, GerryFair, and MultiAccuracy Boost, as a function of varying parameters $n_{bias}$ (top row) and $p_{bias}$ (bottom row). Left: increasing predicted probabilities by $\mu_{sep} = 0.50$. Center: decreasing true probabilities by $\mu_{suf} = 0.50$. Right: base rate difference $\delta = 0.25$, for $\mu_{sep} = \mu_{suf} = 0$.

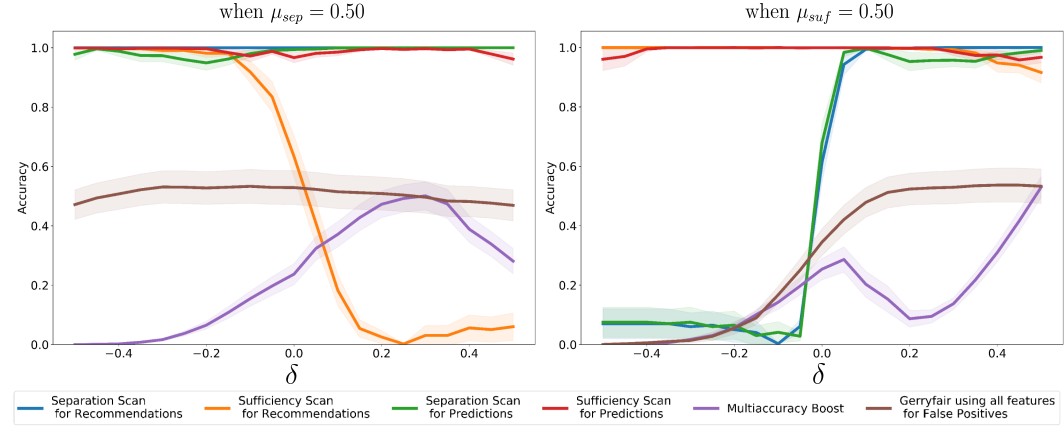

Figure 5: Average accuracy (with 95% CI) for biases injected into subgroup $S_{bias}$ of the protected class, for CBS, GerryFair, and MultiAccuracy Boost, as a function of varying base rate difference $\delta$ between protected and non-protected class for subgroup $S_{bias}$. Left: increasing predicted probabilities by $\mu_{sep} = 0.50$. Right: decreasing true probabilities by $\mu_{suf} = 0.50$.

simulations: (1) varying the number of attribute categories to choose attribute-values from ($n_{bias}$) between 1 and 4, when $p_{bias} = 0.50$; and (2) varying the probability ($p_{bias}$) of an attribute-value being included in $S_{bias}$ between 0 and 1, when $n_{bias} = 2$. The results of these simulations are shown in Figure 4. We observe that, when varying $n_{bias}$, CBS has similar accuracy results to the simulations shown in Figures 1 and 2, with separation scans and sufficiency scan for predictions having higher bias detection accuracy when $\mu_{sep} = 0.50$, and sufficiency scans having higher bias detection accuracy when $\mu_{suf} = 0.50$, as compared to competing methods across all settings of $n_{bias}$. Interestingly, when $\mu_{sep} = 0.50$ and $p_{bias}$ approaches 1 (i.e., more individuals in the protected class are included in $S_{bias}$), GerryFair has improved bias detection accuracy, approaching that of

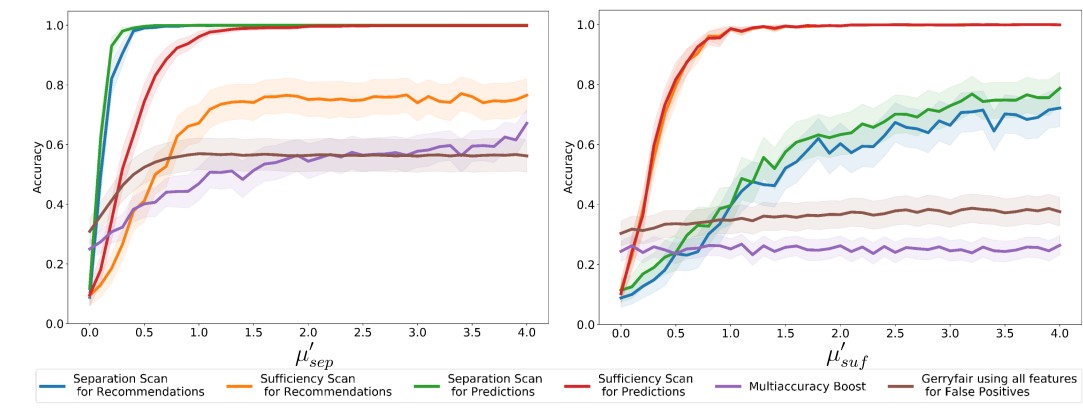

Figure 6: Average accuracy (with 95% CI) as a function of the amount of bias injected into subgroup $S_{bias}$ of the protected class, for four variants of CBS, GerryFair, and MultiAccuracy Boost. Left: increasing predicted log-odds by $\mu'_{sep}$. Right: decreasing true log-odds by $\mu'_{suf}$.

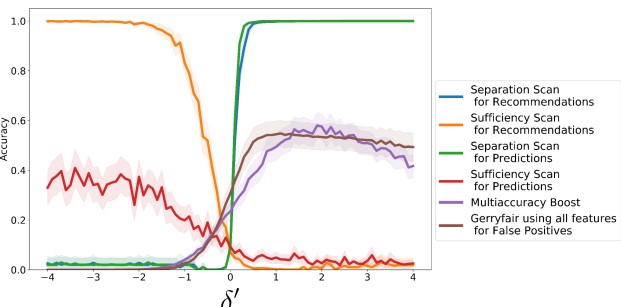

Figure 7: Average accuracy (with 95% CI) as a function of the base rate difference $\delta'$ between protected and non-protected class for subgroup $S_{bias}$, for four variants of CBS, GerryFair, and MultiAccuracy Boost. Note that predictions are well calibrated, $\mu'_{sep} = \mu'_{suf} = 0$.

CBS, but it performs poorly for low values of $p_{bias}$. This suggests that CBS is better at detecting smaller, more subtle subgroups $S_{bias}$ than the competing methods.

Additionally, we investigated the case where we have both an injected bias ($\mu_{sep} = 0.50$ or $\mu_{suf} = 0.50$) and a base rate shift $\delta$ in subgroup $S_{bias}$ for the protected class (Figure 5). We examined the extent to which positive and negative shifts $\delta$ either help or harm the detection accuracy of the various methods. Thus we run two separate sets of experiments with injected bias $\mu_{sep} = 0.50$ and injected bias $\mu_{suf} = 0.50$, while varying the base rate shift $\delta$ from -0.50 to +0.50 for each experiment. A positive $\delta$ means $S_{bias}$ in the protected class has a higher base rate, while a negative $\delta$ means $S_{bias}$ in the protected class has a lower base rate, as compared to $S_{bias}$ in the non-protected class.

In Figure 5, we observe that the detection accuracy of the separation scans increases with $\delta$. This relationship is particularly strong for the experiments with injected bias $\mu_{suf} = 0.50$, in which the separation scans show near-perfect accuracy for large positive $\delta$ and near-zero accuracy for large negative $\delta$. These results are not surprising given the separation scans' sensitivity to positive base rate differences for $S_{bias}$ in the protected class even when no injected bias is present (see Figure 2). We observe that the detection accuracy of the sufficiency scan for recommendations decreases with $\delta$ when $\mu_{sep} = 0.50$, with near-perfect accuracy for large negative $\delta$ and near-zero accuracy for large positive $\delta$. Again, these results are not surprising given the sufficiency scan for recommendations' sensitivity to negative base rate differences for $S_{bias}$ in the protected class even when no injected bias is present (see Figure 2). Finally, we observe that the sufficiency scan for predictions maintains high accuracy for both $\mu_{sep} = 0.50$ and $\mu_{suf} = 0.50$ regardless of the base rate difference $\delta$ for $S_{bias}$ in the protected class.

Lastly, the method we use for injecting bias or shifting the base rate of the affected subgroup $S_{bias}$ in the protected class involves increasing or decreasing the true probabilities and predicted probabilities. Since CBS is designed to detect a constant, additive shift in the true and/or predicted log-odds for a subgroup, $S_{bias}$, in the protected class in comparison to that subgroup in the non-protected class (as shown in the alternative hypotheses contained in Table 2), the simulations are designed to ensure that CBS is robust to injected biases and base rate shifts that do not take the same form as CBS's modeling assumptions. For comparison purposes, we also examine injected biases and base rate shifts represented by shifts in the true and/or predicted log-odds. The resulting Figures 6 and 7 can be directly compared to Figures 1 and 2 respectively. Specifically, we perform the following simulations:

- We increase the predicted log-odds by $\mu'_{sep}$ for $S_{bias}$ in the protected class. Note, this shift is performed prior to the predicted probabilities being drawn for all the data.

- We decrease the true log-odds by $\mu'_{suf}$ for $S_{bias}$ in the protected class. This shift is performed after predicted probabilities have been drawn for all the data. After the true log-odds have been decreased by $\mu'_{suf}$ for $S_{bias}$ in the protected class, outcomes $Y$ are redrawn specifically for the rows of $S_{bias}$ in the protected class.

- We simultaneously shift the true and predicted log-odds by $\delta'$ for $S_{bias}$ in the protected class. Outcomes are redrawn for $S_{bias}$ in the protected class after the shift by $\delta'$ is performed.

In Figure 6, we observe that the injected signals for $\mu'_{sep}$ and $\mu'_{suf}$ (represented as shifts in the predicted and true log-odds respectively) have an effect on CBS's detection accuracy that is nearly identical to the predicted and true probability shifts ($\mu_{sep}$ and $\mu_{suf}$ respectively) shown in Figure 1. Similarly, in Figure 7, we see that the base rate shift created by simultaneously shifting the true and predicted log-odds by $\delta'$ for $S_{bias}$ in the protected class has an effect on CBS's detection accuracy that is nearly identical to the simultaneous shift of the true and predicted probabilities of $S_{bias}$ in the protected class by $\delta$ as shown in Figure 2. Therefore, we can conclude that CBS not only performs well for a constant additive shift in the true and/or predicted log-odds (consistent with its modeling assumptions) but also achieves high detection power for non-additive shifts as shown in Section 4.

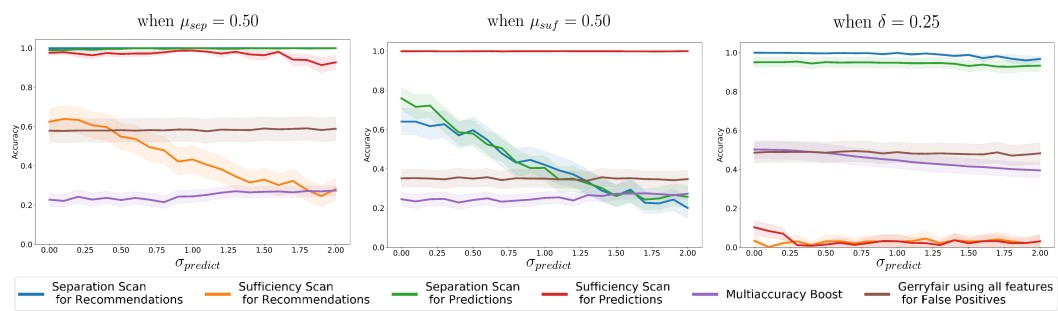

Figure 8: Average accuracy (with 95% CI) for biases and base rate shifts injected into subgroup $S_{bias}$ of the protected class, for CBS, GerryFair, and MultiAccuracy Boost, as a function of varying parameter $\sigma_{predict}$. Left: increasing predicted probabilities by $\mu_{sep} = 0.50$. Center: decreasing true probabilities by $\mu_{suf} = 0.50$. Right: base rate difference $\delta = 0.25$, for $\mu_{sep} = \mu_{suf} = 0$.

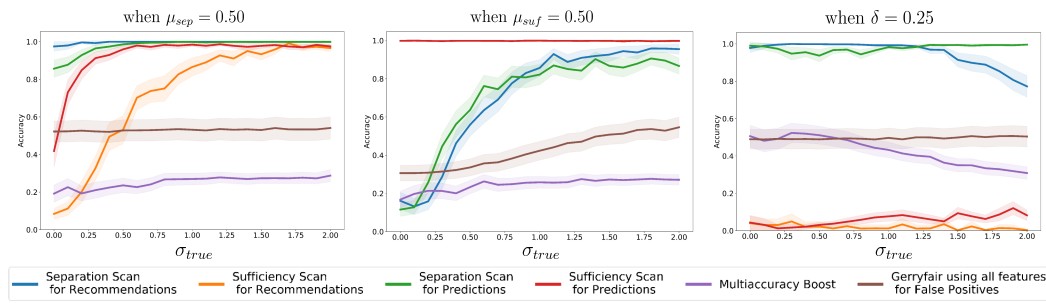

Figure 9: Average accuracy (with 95% CI) for biases and base rate shifts injected into subgroup $S_{bias}$ of the protected class, for CBS, GerryFair, and MultiAccuracy Boost, as a function of varying parameter $\sigma_{true}$. Left: increasing predicted probabilities by $\mu_{sep} = 0.50$. Center: decreasing true probabilities by $\mu_{suf} = 0.50$. Right: base rate difference $\delta = 0.25$, for $\mu_{sep} = \mu_{suf} = 0$.

### B.4 ROBUSTNESS ANALYSES OF EVALUATION SIMULATIONS FOR PARAMETERS $\sigma_{true}$ AND $\sigma_{predict}$

In this section, we examine the robustness of our results in Section 4 by varying the parameters $\sigma_{predict}$ and $\sigma_{true}$ from their default values of 0.2 and 0.6 respectively.

First, we examine the impact of varying $\sigma_{predict}$. Recall that each predicted log-odds is drawn from a Gaussian distribution centered at the true log-odds, with standard deviation $\sigma_{predict}$. Thus $\sigma_{predict}$ can be interpreted as the average amount of random error in the classifier's predictions as compared to the true log-odds values. We run three separate sets of experiments where we alter $S_{bias}$ in the protected class by injecting a bias of $\mu_{sep} = 0.50$, injecting a bias of $\mu_{suf} = 0.50$, and creating a base rate difference of $\delta = 0.25$ respectively, while varying $\sigma_{predict}$ between 0 and 2. Accuracies are averaged over 100 semi-synthetic datasets for each experiment. The experiments where $\mu_{sep} = 0.50$ and $\mu_{suf} = 0.50$ analyze the robustness to $\sigma_{predict}$ of the evaluation simulations for (Q1), whereas the experiments where $\delta = 0.25$ analyze the robustness to $\sigma_{predict}$ of the evaluation simulations for (Q2).

In Figure 8, we observe that large amounts of noise $\sigma_{predict}$ harm the accuracy of the separation scans for injected biases $\mu_{suf} = 0.50$ which shift the true probabilities in subgroup $S_{bias}$ for the protected class. When $\sigma_{predict}$ is large, we see a reduction in accuracy for the sufficiency scan for recommendations for injected biases $\mu_{sep} = 0.50$, which is expected given this scan's initial lower accuracy detection with recommendations with a moderate value of noise in the recommendations.

Second, we examine the impact of varying $\sigma_{true}$. Recall that each individual's true log-odds is a deterministic (linear) function of their covariate values $X_i$ plus a term, $\epsilon_i^{true}$, drawn from a Gaussian distribution centered at 0 with a standard deviation of $\sigma_{true}$. Thus the parameter $\sigma_{true}$ represents the variation between individuals' true log-odds based on characteristics other than the covariate values $X_i$ used by CBS. Moreover, since each individual's predicted log-odds is drawn from a Gaussian distribution centered at the true log-odds, these characteristics are assumed to be known and incorporated into the classifier, thus creating the dependency $Y \not\perp P \mid X$ when $\sigma_{true} > 0$. In other words, $\sigma_{true}$ represents the average amount of signal in the predictions $P$ (for predicting the outcome $Y$) that is not already present in the covariates $X$. We run three separate sets of experiments where we alter $S_{bias}$ in the protected class by injecting a bias of $\mu_{sep} = 0.50$, injecting a bias of $\mu_{suf} = 0.50$, and creating a base rate difference of $\delta = 0.25$ respectively, while varying $\sigma_{true}$ between 0 and 2 for each experiment. Accuracies are averaged over 100 semi-synthetic datasets for each experiment. The experiments where $\mu_{sep} = 0.50$ and $\mu_{suf} = 0.50$ analyze the robustness to $\sigma_{true}$ of the evaluation simulations for (Q1), whereas the experiments where $\delta = 0.25$ analyze the robustness to $\sigma_{true}$ of the evaluation simulations for (Q2).

In Figure 9, we observe that small values of $\sigma_{true}$ harm the accuracy of the separation scans for injected bias $\mu_{suf} = 0.50$ while making them more likely to detect base rate shifts $\delta > 0$ in subgroup $S_{bias}$ for the protected class. Most interestingly, when $\sigma_{true}$ is small, we see a substantial reduction in accuracy for the sufficiency scans for injected bias $\mu_{sep} = 0.50$. This reduced performance for $\sigma_{true} \approx 0$ follows from our argument in Section B.2 above: $\sigma_{true} = 0$ implies $Y \perp P \mid X$, and if we also have no base rate difference between the protected and non-protected classes ($Y \perp A \mid X$), this implies $Y \perp A \mid P, X$. In other words, even if a bias is injected into the predicted probabilities (and recommendations) in subgroup $S_{bias}$ for the protected class, the sufficiency-based definition of fairness is not violated, and thus the injected bias cannot be accurately detected.

### B.5   ESTIMATES OF COMPUTE POWER

For all of the experiments in Section 4, Appendix B.3, and Appendix B.4, with the exception of the experiments displayed in Figure 6 and Figure 7, we used a university's high-performance computing (HPC) services. We completed all these simulations with 100 jobs that used one node, one core (CPU), and 7 GB of memory each. Each of these jobs performed 1,344 CBS runs, and each job was alive for approximately 9 days. To perform the experiments displayed in Figure 6 and Figure 7, as well as additional robustness checks, we used 15 shared, university compute servers running CentOS with 16-64 cores (CPU) and 16-256 GB of memory. Each server performed 15-120 runs of CBS concurrently, and ran for approximately 9 days. We estimate that to run all of the simulations and robust checks (1,344 CBS runs in total) for a single data set using shifts in the predicted and true probabilities for injecting bias and base rate shifts, this would take approximately 9 days. We estimate that to run all of the simulations and robustness checks (1,504 CBS runs in total) for a single data set using shifts in the predicted and true log-odds for injecting bias and base rate shifts, this would take approximately 32.5 hours. Lastly, to run an individual CBS scan for the COMPAS data (150 iterations), it takes on average approximately 90 seconds. A single run of CBS takes a similar runtime for the German Credit Data.

## C   CASE STUDIES APPENDICES

### C.1   CASE STUDY OF COMPAS APPENDICES

#### C.1.1   ADDITIONAL INFORMATION ABOUT PREPROCESSING OF COMPAS DATA

We follow many of the processing decisions made in the initial ProPublica analysis, including removing traffic offenses and defining recidivism as a new arrest within two years of the initial arrest for a defendant (Larson et al., 2016; Larson and Roswell, 2017). After preprocessing the initial data set, we have 6,172 defendants, their gender, race, age (Under 25 or 25+), charge degree (Misdemeanor or Felony), prior offenses (None, 1 to 5, or Over 5), predicted recidivism risk score (1-10), and whether they were re-arrested within two years of the initial arrest.

### C.1.2 Full Results of COMPAS Case Study

Table 4 contains the full set of COMPAS results for CBS.

Table 4: Full table of results for COMPAS case study

| Scan Type | Protected Class Attribute Value | Detected Subgroup | Comparison Subgroup | Score | Observed Rate (Detected) | Observed Rate (Comparison) |
|---|---|---|---|---|---|---|
| Separation Scan for Predictions | Under age 25 | All defendants under age 25 (593) | All defendants age 25+ (2770) | **128.2** | 0.51 | 0.37 |
| | 6+ priors | All defendants with 6+ priors (349) | All defendants with 0-5 priors (3014) | **83.9** | 0.54 | 0.38 |
| | Black | Black male defendants (1168) | Non-Black male defendants (1433) | **42.4** | 0.45 | 0.35 |
| | 1 to 5 priors | Defendants under age 25 with 1 to 5 priors (227) | Defendants under age 25 with 0 or 6+ priors (366) | **3.28** | 0.54 | 0.49 |
| | Felony | White female defendants arrested on felony charges (139) | White female defendants arrested on misdemeanor charges (173) | **2.45** | 0.42 | 0.34 |
| | Female | White female defendants (312) | White male defendants (969) | **1.51** | 0.38 | 0.35 |
| | Male | Asian male defendants (22) | Asian female defendants (1) | 0.63 | 0.30 | 0.22 |
| | Native American | All Native American defendants (6) | All non-Native American defendants (3357) | 0.45 | 0.49 | 0.39 |
| Separation Scan for Recommendations | Under age 25 | All defendants under age 25 (403) | All defendants age 25+ (1583) | **159.3** | 0.53 | 0.25 |
| | 6+ priors | All defendants with 6+ priors (349) | All defendants with 0-5 priors (3014) | **126.9** | 0.66 | 0.26 |
| | Black | Black male defendants (1168) | Non-Black male defendants (1433) | **102.3** | 0.44 | 0.19 |
| | Male | Asian and Hispanic male defendants (286) | Asian and Hispanic female defendants (57) | **22.5** | 0.21 | 0.05 |
| | 1 to 5 priors | Defendants under age 25 with 1 to 5 priors (227) | Defendants under age 25 with 0 or 6+ priors (366) | 12.6 | 0.64 | 0.47 |
| | Female | White female defendants (312) | White male defendants (969) | **12.5** | 0.29 | 0.20 |
| | Felony | White female defendants arrested on felony charges (139) | White female defendants arrested on misdemeanor charges (173) | 9.56 | 0.38 | 0.21 |
| | White | White female defendants under age 25 with no priors (31) | Non-white female defendants under age 25 with no priors (70) | 2.01 | 0.71 | 0.56 |

| | | | | | | |
|---|---|---|---|---|---|---|
| | Misde-meanor | Native American defendants with 1 to 5 priors arrested on misdemeanor charges (2) | Native American defendants with 1 to 5 priors arrested on felony charges (1) | 1.67 | 1.00 | 0.00 |
| | Age 25+ | Asian defendants age 25+ arrested on felony charges (10) | Asian defendants under age 25 arrested on felony charges (1) | 0.74 | 0.20 | 0.00 |
| | Native American | All Native American defendants (6) | All non-Native American defendants (3357) | 0.53 | 0.50 | 0.30 |
| | No priors | All defendants with no priors (2085) | All defendants with 1+ priors (4087) | **111.6** | 0.29 | 0.54 |
| | Age 25+ | Male defendants age 25+ with 0-5 priors (2867) | Male defendants under age 25 with 0-5 priors (1041) | **92.7** | 0.35 | 0.59 |
| | Male | Male Native American defendants of age 25+ (7) | Female Native American defendants of age 25+ (2) | 31.4 | 0.14 | 1.00 |
| | Female | Female defendants under age 25 (246) | Male defendants under age 25 (1101) | **18.7** | 0.38 | 0.60 |
| Sufficiency Scan for Predictions | Misde-meanor | Female defendants arrested on misdemeanor charges (491) | Female defendants arrested on felony charges (684) | 3.51 | 0.26 | 0.41 |
| | Asian | Asian defendants arrested on misdemeanor charges (12) | Non-Asian defendants arrested on misdemeanor charges (2190) | **3.16** | 0.00 | 0.38 |
| | White | White defendants under age 25 (347) | Non-white defendants under age 25 (1000) | 2.36 | 0.49 | 0.58 |
| | Black | Black female defendants (549) | Non-Black female defendants (626) | 2.21 | 0.37 | 0.34 |
| | 1 to 5 priors | Black defendants of age 25+ with 1 to 5 priors (1038) | Black defendants of age 25+ with 0 or 6+ priors (1328) | 2.17 | 0.42 | 0.55 |
| | Hispanic | All Hispanic defendants (509) | All non-Hispanic defendants (5663) | 0.26 | 0.37 | 0.46 |
| | Native American | All Native American defendants (11) | All non-Native American defendants (6161) | 0.14 | 0.45 | 0.46 |
| | Age 25+ | Male defendants of age 25+ with 0-5 priors (772) | Male defendants under age 25 with 0-5 priors (641) | **53.0** | 0.52 | 0.67 |
| | No priors | All defendants with no priors (553) | All defendants with 1+ priors (2198) | **51.0** | 0.46 | 0.67 |
| | 1 to 5 priors | Male defendants of age 25+ with 1 to 5 priors (595) | Male defendants of age 25+ with 0 or 6+ priors (981) | **26.8** | 0.54 | 0.70 |

|  |  | Detected subgroup $S^*$ | Comparison subgroup |  |  |  |
| --- | --- | --- | --- | --- | --- | --- |
|  | Male | Male Native American defendants of age 25+ (4) | Female Native American defendants of age 25+ (2) | 14.1 | 0.25 | 1.00 |
| Sufficiency Scan for Recommendations | Female | Female defendants under age 25 (167) | Male defendants under age 25 (699) | **13.2** | 0.44 | 0.68 |
|  | Misdemeanor | All defendants on misdemeanor charges (736) | All defendants on felony charges (2015) | **10.7** | 0.55 | 0.66 |
|  | Hispanic | All Hispanic defendants (141) | All non-Hispanic defendants (2610) | **2.48** | 0.56 | 0.63 |
|  | 6+ priors | Asian defendants with 6+ priors (1) | Asian defendants with 0-5 priors (6) | 0.42 | 0.00 | 0.83 |
|  | White | White female defendants under age 25 (57) | Non-white female defendants under age 25 (110) | 0.41 | 0.39 | 0.47 |
|  | Black | Black defendants of age 25+ with 0-5 priors (581) | Non-Black defendants of age 25+ with 0-5 priors (404) | 0.37 | 0.50 | 0.52 |
|  | Asian | Asian defendants with 6+ priors (1) | Non-Asian defendants with 6+ priors (965) | 0.11 | 0.00 | 0.76 |

Each of the four variants of CBS was run using each observed attribute value as the protected class. Detected subgroup $S^*$ of the protected class and corresponding (comparison) subgroup of the non-protected class; numbers of defendants for each subgroup are shown in parentheses. All runs with log-likelihood ratio score $F(S^*) > 0$ are shown, sorted in descending order by score for each method. Separation scan for predictions: "observed rate" is average predicted probability of reoffending, $\mathbb{E}[P_i]$, for defendants who did not reoffend ($Y_i = 0$). Separation scan for recommendations: "observed rate" is false positive rate, i.e., proportion of individuals predicted as "high risk" ($P_{i,bin} = 1$) for defendants who did not reoffend ($Y_i = 0$). Sufficiency scan for predictions: "observed rate" is proportion of reoffending individuals ($Y_i = 1$), controlling for predicted risk. Sufficiency scan for recommendations: "observed rate" is positive predictive value, i.e., proportion of reoffending individuals ($Y_i = 1$) for defendants who were predicted as "high risk" ($P_{i,bin} = 1$). Bolded scores are statistically significant with p-value <.05 measured by permutation testing, as described in Appendix A.3.

### C.1.3 GENDER BIAS IN COMPAS

**Gender bias in COMPAS.** While male and female defendants have equal false positive rates overall, separation scan for recommendations detects a statistically significant gender bias: non-reoffending white female defendants have a higher false positive rate than non-reoffending white male defendants (0.29 vs 0.20). Separation scan for predictions detects the same gender bias but to a lesser degree: non-reoffending white females have an expected risk of 0.38, compared to non-reoffending white males with an expected risk of 0.35. Sufficiency scans for both recommendations and predictions detect a statistically significant over-estimation bias for females under the age of 25. 44% of females under the age of 25 who are flagged as "high-risk" by COMPAS reoffend, as compared to a 68% recidivism rate for males under the age of 25 who are flagged as "high-risk" by COMPAS. For both sufficiency and separation scans, thresholding the risk scores to create recommendations results in larger deviations between the subgroups of females and males found by the scans, thereby exacerbating the underlying biases present in the COMPAS risk scores that adversely impact white female defendants and younger female defendants respectively. Lastly, separation scan for recommendations finds that non-reoffending Asian and Hispanic male defendants have a statistically significant higher false positive rate of being flagged as high-risk (0.21) in comparison to non-reoffending Asian and Hispanic female defendants (0.05) showing that the COMPAS risk scores have intersectional gender biases (in the form of separation violations) that adversely impact different subgroups of male and female defendants.

### C.1.4 CONSIDERATIONS AND LIMITATIONS OF COMPAS DATA AND FAIRNESS DEFINITIONS IN OUR COMPAS CASE STUDY

Following the initial investigation by ProPublica about fairness issues in COMPAS risk predictions (Angwin et al., 2016b), ProPublica's COMPAS dataset has been used as a benchmark in the fairness literature. While we use the COMPAS data because of its familiarity and supporting research, we also note the value of alternative framings of the evaluation of automated decision support tools in the criminal justice systems, such as examining the risks that the system poses to defendants rather than the risk of the defendants to public safety (Mitchell et al., 2021; Meyer et al., 2022; Green, 2020). Beyond the implications of the traditional framing of pre-trial risk assessment tools, there have been specific critiques of the COMPAS data that range from questioning the accuracy of the sensitive attributes (specifically race), noting missing features in the ProPublica dataset that the COMPAS creators claim are important for score calculations, and most importantly, a lack of evaluation of the biases that exist in the outcome variable of whether a defendant is rearrested within two years of arrest (Fabris et al., 2022). Given that certain types of individuals are arrested at a higher rate than others, the outcome variable of re-arrest most likely under- and over-represents certain subpopulations of defendants.

In our COMPAS case study, for the separation scans, we search for subgroups of the protected class with the most significant *increase*, either in the probabilistic predictions or in the probability that the binarized recommendation equals 1, conditional on the defendant's covariates. Moreover, we perform value-conditional scans, focusing specifically on the subset of defendants who did not reoffend ($Y_i = 0$). For the separation scan for recommendations, this results in CBS detecting subgroups of the protected class for whom the *false positive rate* is most significantly increased. For the sufficiency scans, we search for subgroups of the protected class with the most significant *decrease* in the observed rate of reoffending, conditional on the defendant's covariates and their COMPAS prediction or recommendation. For the sufficiency scan for recommendations, we also perform a value-conditional scan. We focus specifically on the subset of defendants who were predicted to be "high risk" by COMPAS ($P_{i,bin} = 1$) because this labeling could negatively impact the defendant, e.g., by decreasing their likelihood of pre-trial release. This results in CBS detecting subgroups of the protected class for whom the *false discovery rate* is most significantly increased. These fairness definitions neglect bias detection for defendants who reoffend (for separation scans) and defendants who are not flagged as high-risk (for sufficiency scan for recommendations). These choices were made to ensure our ability to verify our findings based on previous research on COMPAS which commonly focus on similar fairness violations to those used in our case study. With that said, we strongly encourage auditing for predictive biases that affect reoffending defendants and low-risk defendants as well, if using CBS to audit an algorithmic risk assessment tool in practice. For example, auditing for the increased probability of being flagged as high-risk for reoffending defendants could help to uncover subpopulations that are over-prosecuted in comparison to other

populations of reoffending defendants. Therefore, expanding the fairness definitions used to audit pre-trial risk assessment tools for biases could have beneficial findings.

### C.1.5 DISCUSSION OF COMPAS RESULTS FOR BENCHMARK METHODOLOGIES

Our evaluation of CBS, GerryFair, and MultiAccuracy Boost (Section 4) uses semi-synthetic data that maintains the covariate distribution of COMPAS. The evaluation simulations follow a framework that employs certain generative assumptions for injecting bias into subgroups. The limitations of these generative assumptions used in our framework are discussed in detail in Section 6. In this Appendix, we provide the results of the benchmark methodologies (GerryFair and MultiAccuracy Boost) run on the original COMPAS data, and compare these results to the CBS results for the COMPAS case study in Section 5. We include these results to highlight the differences between CBS and the benchmark methodologies on a non-synthetic dataset, showing the benefits of CBS in a setting without the generative model assumptions used in Section 4.

We ran GerryFair and MultiAccuracy Boost using the same COMPAS data, preprocessing steps, and setup described in Section 5 and Appendix C.1.1. We report two sets of results: (1) the results of these methodologies with their out-of-the-box settings; and (2) the results when using the minimum modifications needed to adapt these methods for under-estimation and over-estimation bias, described in Appendix B.1. We include both of these results to display the methodologies' default functionality, which we assume is the intended setting for practitioners, and to obtain a set of results for COMPAS data that can be used to contextualize the differences between these benchmark methodologies and CBS in a real-world setting. GerryFair and MultiAccuracy Boost provide demonstration code that uses probabilities as the predictive output to be audited, and therefore we use the same $P_i$ calculated for each defendant based on their COMPAS risk score, as described in Section 5.

*GerryFair Results:* When running GerryFair to detect intersectional biases in false positive rates, with race, sex, and the indicator variable of whether defendants are under the age of 25 marked as sensitive attributes, the detected subgroup consists of all defendants aged 25+ who are not Black or Native American. This subgroup is systematically *advantaged* rather than disadvantaged: non-reoffending defendants in the detected subgroup have an average predicted risk $\mathbb{E}(P \,|\, Y = 0) = 0.32$, while non-reoffending defendants not included in this subgroup have an average predicted risk $\mathbb{E}(P | Y = 0) = 0.45$. When modified to perform a directional scan, and searching for a systematically disadvantaged subgroup, GerryFair detects a subpopulation consisting of three distinct, marginal groups—all defendants under 25, all Black defendants, and all Native American defendants—rather than an intersectional or contextual subgroup.

*MultiAccuracy Boost Results:* MultiAccuracy Boost chooses between three partitions of data on each iteration of the algorithm, where the chosen partition has its probabilities adjusted. When running MultiAccuracy Boost with its default settings on COMPAS, the highest scoring partition is found on the first iteration. This partition consists of all defendants in the initial iteration that had higher probabilities ($P > 0.50$), and therefore each of those defendants' probabilities gets adjusted depending on their custom residual heuristic (see Appendix B.1). Given that there are large overlaps in the covariate spaces of the partition that gets its predictions adjusted and the other partitions, the best way to describe this partition's covariate space is based on the coefficients of the classifier used to model the custom residual heuristic, as described in Appendix B.1, where larger values contribute to larger adjustments needed to the probabilities of the defendants in the detected subgroup. The factors that are associated with defendants in this partition needing larger adjustments to their probabilities include defendants with no priors and Hispanic defendants. We note that this algorithm is stochastic, but these covariates consistently show a positive association with larger values of the adjustment heuristic.

When running MultiAccuracy Boost using the modifications described in Appendix B.1 to detect directional bias, the highest scoring partition is found on the first iteration of the algorithm. We find that the factors that estimate the level of adjustments needed to the defendant's probabilities include defendants with no priors, Hispanic and Female defendants, defendants of age 25+, and defendants arrested on misdemeanor charges.

*Discussion:* There are several takeaways to highlight about the results of GerryFair and MultiAccuracy Boost for COMPAS:

- GerryFair's original implementation of its auditor does not allow the user to select between detection of over-estimation bias and detection of under-estimation bias. This results in a detected subgroup of non-reoffending defendants that is advantaged rather than disadvantaged, benefiting from lower predicted risk.

- With our modification to detect directional bias, GerryFair finds a large subpopulation consisting of all Black defendants, all Native American defendants, and all defendants under the age of 25. The results of CBS for separation scans for predictions (Appendix C.1.2) show some similarities with GerryFair's results – that is, for each of the three protected classes included in GerryFair's results, the subgroups detected by CBS within the protected class also have positive scores. The major distinction is that GerryFair is *not* detecting intersectional or contextual subgroups within the protected class, such as the subgroup of Black males detected by CBS. In contrast, CBS identifies that non-reoffending Black male defendants have a higher predicted risk compared to non-reoffending non-Black male defendants, and that this identified racial disparity is more significant than the disparity between all non-reoffending Black defendants and all non-reoffending non-Black defendants.

- More generally, GerryFair appears to lack the flexibility of CBS to specify a single protected class and search for intersectional or contextual subgroups within that protected class for whom bias is present. In the given example, it identifies some individuals using characteristics unrelated to race, and the marginal subgroups of all Black defendants who did not reoffend and all Native American defendants who did not reoffend respectively. This is consistent with our evaluation results in Section 4, in which GerryFair was able to reliably detect marginal biases (for simulation parameter $p_{bias} = 1$) but had low power to detect smaller, more subtle subgroup biases.

- The results of MultiAccuracy Boost suggest that while MultiAccuracy Boost provides a black-box auditor tool, its auditor does not provide interpretable results. This is because the algorithm forms subgroups based only on prediction thresholding, which results in these subgroups having overlapping covariate spaces. This, in combination with the method's inability to audit for specific biases for specified protected class attributes, results in the algorithm neglecting to find important intersectional biases. This is evident from the factors that describe over-estimation bias being defendants of age 25+, defendants with no priors, Hispanic and female defendants, which somewhat aligns with CBS's results for sufficiency scan for predictions for COMPAS, but does not have the capabilities to also find more subtle biases such as the subgroup of Asian defendants arrested on misdemeanor charges affected by over-estimation bias.

In summary, we believe that the above results demonstrate the advantages of CBS as compared to competing methods, as an auditor for detecting intersectional and contextual biases in a real-world context.

## C.2 CASE STUDY OF GERMAN CREDIT DATA

We present the results of using CBS to audit for predictive bias in algorithmically-generated risk scores for customers in the German Credit Data (Hofmann, 1994). This dataset contains information about 1,000 customers from a German financial institution. Each row of the dataset represents a customer. For each customer, various pieces of demographic, socioeconomic, and financial information are available, as well as a label generated by the financial institution indicating whether each customer is a "good" (trustworthy for credit) or "bad" (untrustworthy for credit) customer. This dataset is often used in the fair machine learning literature to evaluate the predictive bias of models estimating credit risk. This is also the context we assume for these data. We include these appendices to demonstrate the use of CBS for an additional dataset. This case study also provides an example of running CBS on a notably smaller data set: the German Credit Data is less than one sixth of the size of the COMPAS data in terms of rows. Below we provide the same set of results as those shown for COMPAS above.

### C.2.1 PREPROCESSING OF GERMAN CREDIT DATA

We use a publicly available version of the German Credit Data that has mapped the keys in the original Statlog data file to their decoded categories (Datahub.io, 2019).

We follow the feature selection and preprocessing methods documented in Kamiran and Calders (2009), which is one of the first publications that used these data for fair machine learning research. For each customer, we use the following information:

- Whether the customer is under age 26 or age 26+.

- Whether the customer owns, rents, or lives in their housing for free.

- The customer's gender and marital status. These were initially coded as one variable. For CBS we create two separate categories for gender and marital status. Additionally, we create two high-level categories for marital status: single or married/separated/divorced/widowed (i.e., "non-single").

- The customer's credit history. We recode this category to the following schema: previously delayed credit/ critical credit/other existing credit or no credit/all credit paid. This involved combining the "no credit/ all credit paid", "all paid", and "existing credit paid" categories because of their overlap. Additionally, we combine previously delayed credit and critical credit/ other existing credit categories because of a lack of clear differences between the categories. The main motivation of these simplifications was to ensure that each category was not overlapping and thus to increase interpretability. We note that there is a lack of granularity specifying if the customer has never had credit before or has no credit because they have paid off all their previous credit for most of the customers in the data set. This is why we see a correlation between customers being labeled as untrustworthy for credit and customers in the category of "no credit/all paid".

- Whether a customer is considered a trustworthy or untrustworthy customer for credit by the financial institution. An untrustworthy customer is coded as a positive outcome and a trustworthy customer as a negative outcome for consistency with the COMPAS case study's outcome label.

Unlike COMPAS, which provides both an algorithmically-generated risk score and an observed outcome for each row, the German Credit Data only provides the label of whether a customer is trustworthy or untrustworthy for credit, which is commonly used as an outcome variable. To produce the equivalent of an algorithmically-generated risk score for each customer, which we will subsequently audit for predictive bias, we train a logistic regression model using credit history, age (under 26 or age 26+), and housing ownership as predictors and the binary indicator of whether the customer is trustworthy or untrustworthy for credit as the label. We use this model to produce the predicted probability that each customer is untrustworthy for credit. These predicted probabilities, and the corresponding binarized recommendations as to whether each customer is predicted high-risk or low-risk of being untrustworthy for credit, are the predictive risk scores that we audit with CBS. This modeling approach is an example of "fairness through unawareness" because it does not use the two sensitive attributes (gender and marital status) as predictors in training to produce its predictions and recommendations. We will examine whether the predictions and recommendations produced by this model still contain predictive biases, as identified by CBS.

### C.2.2 SCANS FOR THE GERMAN CREDIT DATA

We preprocessed the outcome variable (whether a customer is trustworthy or untrustworthy for credit) in a similar fashion to the COMPAS outcome variable. A positive outcome represents a less desirable real-world result. For the German Credit Data, this means that a positive outcome represents an observed untrustworthy customer for credit. Therefore, we run the same scans in terms of conditional variables and direction for the German Credit Data that we ran for COMPAS. For the separation scans, we detect positive deviations for the protected class attribute in $\mathbb{E}(P \mid Y = 0, X)$ and $\Pr(P_{bin} = 1 \mid Y = 0, X)$, i.e., increase in average predicted risk for trustworthy customers and increase in FPR (probability of being predicted high-risk for trustworthy customers), respectively. For the sufficiency scans, we detect a negative deviation for the protected class in $\Pr(Y = 1 \mid P, X)$ and $\Pr(Y = 1 \mid P_{bin} = 1, X)$, i.e., decreased probability of being an untrustworthy customer conditional on predicted risk and conditional on being predicted as high-risk, respectively. For the separation and sufficiency scans for recommendations, we threshold the probability risk-scores by 0.5 to construct recommendations: $P_{bin} = \mathbf{1}\{P \geq 0.5\}$. Given the smaller dataset size (as compared to COMPAS) and highly-correlated predictor variables, we found that logistic regression was inadequate for computing propensity scores and for the outcome model (predicting the probabilities $\hat{I}$ using data

from the non-protected class). Thus we use a more flexible model– a gradient boosting classifier with Platt scaling – to ensure that our predictions are well-calibrated when computing propensity scores and when estimating $\hat{I}$. All scans were run for 500 iterations with a penalty equal to 1.

### C.2.3    Results of German Credit Data Case Study

Table 5 contains the full set of German Credit Data results for CBS. We observe that the statistically significant biases detected by separation scans are those corresponding to subpopulations with higher base rates (i.e., higher probability of being labeled "untrustworthy" for credit): customers with all paid or no previous credit, younger customers, and customers who have free housing or rent their housing. For sufficiency scans, we detect only a single statistically significant bias: conditional on predicted risk, older female customers with all paid or no previous credit who own their housing are significantly less likely to be labeled as "untrustworthy" than older female customers with all paid or no previous credit who rent or have free housing.

As described in Appendix C.2.1, we purposely excluded the gender and marital status features when modeling the risk scores. Since the exclusion of sensitive features alone does not guarantee that a model will produce predictions without predictive biases, we examine gender biases detected in the logistic regression model's risk scores. It is notable that a sufficiency scan for recommendations identifies a subgroup of female customers who own or rent their housing, have critical, previously delayed, or other existing credit, and are aged 26 or older who are flagged as high-risk for credit. This subgroup has a lower rate of being untrustworthy for credit (0.12) compared to the equivalent group of male customers predicted as high-risk for credit, where the rate of being untrustworthy for credit is 0.19. This scan additionally detects that male customers who have free housing and are predicted as high-risk have a lower rate of being untrustworthy for credit (0.37) as compared to female customers who have free housing and are predicted as high-risk (0.58). Although neither of these detected subgroups is statistically significant, they do represent deviations, in the form of miscalibrated predictions, that disadvantage a subgroup of customers based on their gender as compared to the opposite gender. This suggests that removing gender and marital status as predictors may not be sufficient to fully remove gender-related subgroup biases in the model predictions.

Table 5: Full table of results for German Credit Data case study

| Scan Type | Protected Class Attribute Value | Detected Subgroup | Comparison Subgroup | Score | Observed Rate (Detected) | Observed Rate (Comparison) |
|---|---|---|---|---|---|---|
| Separation Scan for Predictions | All paid or no previous credit | All customers with all paid or no previous credit (397) | All customers with critical, previously delayed or other existing credit (303) | **86.5** | 0.35 | 0.20 |
| | Under age 26 | All customers under age 26 (110) | All customers of age 26+ (590) | **13.5** | 0.41 | 0.26 |
| | Free housing | All customers who have free housing (64) | All customers who own or rent their housing (636) | **12.9** | 0.39 | 0.28 |
| | Rent their housing | All customers who rent their housing (109) | All customers who own or have free housing (591) | **5.62** | 0.38 | 0.27 |
| Separation Scan for Recommendations | Single | Single customers under age 26 who have free housing (2) | Non-single customers under age 26 who have free housing (1) | 9.19 | 1.00 | 0.00 |
| | Male | Male customers under age 26 who have free housing (2) | Female customers under age 26 who have free housing (1) | 8.39 | 1.00 | 0.00 |
| | Free housing | Customers under age 26 who have free housing (3) | Customers under age 26 who own or rent their housing (107) | **3.02** | 0.67 | 0.32 |
| | Non-single | Non-single customers who rent their housing (74) | Single customers who rent their housing (35) | 2.39 | 0.42 | 0.09 |
| | Female | All female customers (201) | All male customers (499) | 0.08 | 0.11 | 0.03 |
| | Own their housing | Female customers of age 26+ with all paid or no previous credit who own their housing (93) | Female customers of age 26+ with all paid or no previous credit who rent or have free housing (42) | **81.2** | 0.33 | 0.50 |
| | Age 26+ | Single customers who own their housing of age 26+ (366) | Single customers who own their housing under age 26 (42) | 42.4 | 0.22 | 0.36 |
| | Critical, previously delayed or other existing credit | Customers who own their housing of age 26+ with critical, previously delayed or other existing credit (267) | Customers who own their housing of age 26+ with all paid or no previous credit who own their housing (340) | 8.80 | 0.16 | 0.29 |

| Method | Protected class | Detected subgroup $S^*$ | Comparison subgroup | $F(S^*)$ | | |
|---|---|---|---|---|---|---|
| | Female | Female customers who own or rent their housing with critical, previously delayed or other existing credit of age 26+ (66) | Male customers who own or rent their housing with critical, previously delayed or other existing credit of age 26+ (234) | 7.31 | 0.12 | 0.19 |
| Sufficiency Scan for Predictions | Male | Male customers who have free housing (89) | Female customers who have free housing (19) | 6.23 | 0.37 | 0.58 |
| | Single | Single customers who have free housing (85) | Non-single customers who have free housing (23) | 4.92 | 0.38 | 0.52 |
| | Rent their housing | Female customers who rent their housing (95) | Female customers who own or have free housing (215) | 1.91 | 0.41 | 0.33 |
| | All paid or no previous credit | Single customers of age 26+ who own their housing with all paid or no previous credit (189) | Single customers of age 26+ who own their housing with critical, previously delayed or other existing credit (177) | 1.55 | 0.28 | 0.15 |
| | Under age 26 | All customers under age 26 (190) | All customers of age 26+ (810) | 0.07 | 0.42 | 0.27 |
| | Non-single | All non-single customers (56) | All single customers (12) | 0.54 | 0.45 | 0.58 |
| Sufficiency Scan for Recommendations | Male | All male customers (24) | All female customers (44) | 0.02 | 0.46 | 0.48 |

Each of the four variants of CBS was run using each observed attribute value as the protected class. Detected subgroup $S^*$ of the protected class and corresponding (comparison) subgroup of the non-protected class; numbers of customers for each subgroup are shown in parentheses. All runs with log-likelihood ratio score $F(S^*) > 0$ are shown, sorted in descending order by score for each method. Separation scan for predictions: "observed rate" is average predicted risk, $\mathbb{E}[P_i]$, for customers who are trustworthy for credit ($Y_i = 0$). Separation scan for recommendations: "observed rate" is false positive rate, i.e., proportion of individuals predicted as "high-risk" ($P_{i,bin} = 1$) for customers who are trustworthy for credit ($Y_i = 0$). Sufficiency scan for predictions: "observed rate" is proportion of untrustworthy customers for credit ($Y_i = 1$), controlling for predicted risk. Sufficiency scan for recommendations: "observed rate" is positive predictive value, i.e., proportion of untrustworthy customers ($Y_i = 1$) for customers who were predicted as "high-risk" ($P_{i,bin} = 1$). Some subgroups are not included for binary sufficiency and binary separation scans because the limited range of the predicted risk score prevented auditing with CBS. We note that the three lowest-scoring subgroups for sufficiency scan for predictions had higher observed rates in the detected group vs. comparison group. These observed rates were still lower than expected, resulting in small but non-zero scores, given the systematic differences in other predictors between protected and non-protected class. Bolded scores are statistically significant with p-value <.05 measured by permutation testing, as described in Appendix A.3. "Non-single" is short for the marital status attribute "Married/divorced/separated/widowed".

### C.2.4    German Credit Data Results for Benchmark Methodologies

We use the same setup described in Appendix C.1.5 for running the benchmark methodologies with their default settings and with the modifications to account for directional bias. Additionally, we use the same data and risk scores described in the other sections of Appendix C.2.

*GerryFair Results:* When running GerryFair with its default settings of detecting positive or negative deviations in the false positive rate in comparison to the global false positive rate with marital status and gender marked as sensitive attributes, GerryFair detects a subgroup of single male customers with a slightly decreased average predicted risk for credit of 0.27 for trustworthy customers in comparison to the global average predicted risk score of 0.29 for trustworthy customers. This is a negative deviation in the false positive rate. The German Credit dataset contains no single females. When running GerryFair to detect positive deviations in the false positive rate, it detects a subgroup of credit-trustworthy married/divorced/separated/widowed customers (i.e., "non-single") who have a slightly increased average predicted risk of 0.30 in comparison to the global expected risk score of 0.29 for all trustworthy customers.

*MultiAccuracy Boost Results:* The MultiAccuracy Boost results, both for its default settings and when accounting for over-estimation bias, found no noteworthy associations between the coefficients of the predictors used to estimate the custom residual heuristic used in MultiAccuracy Boost. This further substantiates our claim that MultiAccuracy Boost does not have the capabilities to be easily used as an auditing tool for subgroup predictive biases.

## D    Additional Related Work

Our discussion of related work in Section 2, and our empirical comparisons in Section 4, are focused on the foundational papers in the machine learning literature on *auditing classifiers for intersectional and subgroup biases*, e.g., Kearns et al. (2018) and Kim et al. (2019a). These papers are used as benchmarks for our method.

There is other research for subgroup bias auditing which is not directly comparable to CBS. For example, Chouldechova and G'Sell (2017) use a recursive partitioning algorithm to find subgroups where the false positive rate disparity between individuals in the protected and non-protected class differs between two predictive models. In addition to this framework providing limited fairness metrics for auditing, this work is formulated to measure pairwise disparities between two models' predictive performance, whereas CBS separately audits each predictive model's results, making this work ill-suited as a benchmark for CBS.

Additionally, we reference the concept of intersectionality in our main paper, which has a rich history (Crenshaw, 1991a;b; Collins, 2008). Given the importance of intersectional biases, we provide concise resources for the original conceptualizations of 'intersectionality'. In the sociology literature, intersectionality theory (Crenshaw, 1991a;b; Collins, 2008) describes how individuals' different social positions and identities interact to influence their social experiences, actions, and outcomes. In particular, an individual at the intersection of several minoritized groups may be impacted by multiple historical and continuing systems of power and oppression (structural racism, sexism, income and wealth disparities, etc.).

Several recent quantitative research papers (Bose and Hamilton, 2019; Foulds et al., 2020; Subramanian et al., 2021) have proposed methods for *learning fair classifiers* (as opposed to auditing classifiers) with respect to intersectional and/or contextual biases. In the machine learning literature, Bose and Hamilton (2019) use filtered embeddings to train debiased graph embeddings; Foulds et al. (2020) propose new definitions of intersectional bias and use regularization to train fair classifiers; and Subramanian et al. (2021) propose a classifier trained with bias-constraints and also extend a post-hoc debiasing method called iterative nullspace projection (INLP) to address intersectional bias. As noted above, Bose and Hamilton (2019), Foulds et al. (2020), and Subramanian et al. (2021) focus on learning fair classifiers as opposed to auditing classifiers. While INLP could conceivably be adapted for auditing given its similarity to the iterative postprocessing method used by MultiAccuracy Boost discussed in Section 2 and used as a benchmark, this approach does not find the subgroup with the *most* systematic bias on any given iteration, a significant and novel contribution of Conditional Bias Scan.

We present a novel subgroup discovery algorithm to search for predictive bias. Subgroup discovery is a rich research domain. Herrera et al. (2011) provide a comprehensive overview of subgroup discovery, covering various fundamental topics including a sampling of search algorithms and quality measurements. Klösgen (1999) provides a condensed and select overview of the topic of subgroup discovery. Lastly, Leman et al. (2008) present a framework for multi-target attribute subgroup discovery. While this work is significantly different from CBS regarding framing, quality measurements, search algorithms, etc., it provides a useful overview of various considerations of subgroup discovery pertaining to a model's outputs for a given data distribution.

# E    BROADER SCOPE OF IMPACT

CBS is, to our knowledge, the first auditing tool that can answer whether there are intersectional biases that adversely impact a given protected class or any subgroup of that protected class. The other tools mentioned in Section 2 and Appendix D either do not account for directional bias, do not audit for predictive biases impacting a given protected class or subgroup of that protected class, or were not designed for auditing a single model. Given the ultimate objective of understanding the full scope of predictive biases that a model produces for *all* the sensitive subpopulations of a given target population, there is the need for expanded measurements of predictive bias and improved methods for searching for these biases within all sensitive subpopulations that could be adversely affected by predictive bias. Without auditing tools that can robustly search for these biases, any predictive bias definition will be limited to evaluating a limited, static set of subpopulations, and there will presumably be some form of intersectional or contextual bias that goes undetected. Practitioners can use CBS to determine if a model's predictions are biased for *any* subgroup of a protected class, therefore can identify intersectional and contextual biases that impact any subpopulation defined by protected class membership. We demonstrate this with our case studies of the COMPAS risk scores (Section 5 and Appendix C.1) and German Credit Data (Appendix C.2). Therefore, CBS is an important step toward understanding the full scope of predictive biases a model might produce. Ultimately, this methodology can play a role in ensuring that machine learning models used in socio-technical settings are not exacerbating societal harms.

Since CBS is solely an auditing methodology, it presents less risk than a method that intends to mitigate predictive biases. With that said, auditing tools for predictive models can inadvertently suggest that the most beneficial course of action is to correct predictive biases. As discussed in Section 6, predictive biases could exist for a variety of reasons, and often align with larger societal disparities. Understanding and mitigating biases in predictive models are important goals, but do not eliminate the pressing need to address the societal disparities which are the root causes of these biases.

We use the COMPAS data as one of our case studies for CBS. In Appendix C.1.4 we discuss various issues pertaining to the COMPAS data and its use in fair machine learning research, as well as exploring the implications of the fairness definitions we chose for the COMPAS case study. Our use of COMPAS was motivated by easily available data to verify our auditing methodology. We have no intention of endorsing, solidifying or normalizing the use of risk assessment scores in arraignment settings. In Appendix C.1.4, we provide references to research critical of the current framing of risk assessment tools in arraignment courts, and alternative framings for risk assessments pertaining to criminal justice, such as assessing the risk posed to defendants because of interactions with the criminal justice system.

