# OpenReview forum: "Auditing Predictive Models for Intersectional Biases"
_ICLR.cc/2025/Conference — ICLR 2025 Conference Withdrawn Submission_

### Official Review · Reviewer_rwk3 · 2024-10-22

**Soundness:** 2
**Presentation:** 2
**Contribution:** 3
**Rating:** 5
**Confidence:** 2

**Summary:**

The authors address the challenge of auditing machine learning models for intersectional biases (fairness gerrymandering). They introduce a methodology called Conditional Bias Scan (CBS) for detecting biases that affect specific subgroups, which may arise from intersectional factors (membership in two or more protected classes) or contextual factors (decision situations). The CBS methodology involves four stages: (1) initializing the event variable I, protected class A, covariates X, and conditional variable C based on the input parameters and chosen fairness definition; (2) estimating the expected value of I under the null hypothesis; (3) using a multidimensional subset scan to identify subgroups that systematically deviate from the expected values computed in step (2) and selecting the most significant ones; and (4) assessing the statistical significance of the detected subgroups. The paper includes a comprehensive experimental evaluation to validate the approach.

**Strengths:**

1. The paper provides a framework for auditing intersectional biases, a crucial area often overlooked in fairness assessments (detection of gerrymandering).
2. The proposed method can accommodate different group fairness metrics and can effectively scan numerous subgroups.

**Weaknesses:**

1. The reliability of the estimation of expectations I under the null hypothesis depends on having well-specified models for estimating the propensity scores of the protected class.
2. The paper is quite dense and challenging to follow. It would benefit from providing more intuitive explanations or examples to illustrate why the overall method is effective in real-world scenarios. This would help readers better understand the practical implications and the rationale behind the approach.

**Questions:**

1. How does CBS handle scenarios with continuous covariates without discretization, and how does it address the potential loss of valuable information during this process?
2. What are the implications of using different models for estimating conditional expectations on the detection of biases? Beyond modeling the COMPAS data, is there additional evidence of real-world datasets to support the effectiveness of this method? See also weakness 2.
3. Could you clarify the distinction between auditing the COMPAS dataset and the model itself? The explanation in Section 5 is not entirely clear.
4. Regarding Figure 3: Does the framework detect the entire group of defendants under the age of 25? What is the role of the conditional variable C in the null hypothesis in this case?
5. Consider making the tables more consistent in terms of formatting and presentation for easier comparison.

---

### Official Review · Reviewer_mxzy · 2024-10-28

**Soundness:** 2
**Presentation:** 2
**Contribution:** 2
**Rating:** 5
**Confidence:** 3

**Summary:**

The paper presents a novel approach to auditing and detecting fairness biases in predictive models. The method, called Conditional Bias Scan (CBS), allows for identifying the subgroup with the most significant bias and comparing it with the equivalent subgroup in the non-protected class. Empirical evaluation suggests the effectiveness of the approach.

**Strengths:**

The main strengths of the paper are:

S1) the motivation of the methodology is relevant, as identifying intersectional biases (in a tractable manner) is an open issue in the fairness literature;

S2) the empirical evaluation supports the effectiveness of the method;

S3) the algorithmic procedure for detecting the most significant subgroup seems novel.

**Weaknesses:**

The main shortcomings of the current version of the paper are:


W1) I think a few arguments should be taken into account and need further clarification:
* in [Ruggieri et al., 2023], the authors show that algorithmic fairness objectives are not compositional, i.e., even if the classifier is fair on some of the regions of the input space, due to the emergence of Yule’s effect, the overall system is not necessarily fair. This could hinder CBS's ability to evaluate the overall fairness of the system.
* in lines 183-184, the authors consider propensity score estimates for $Pr(A=1|X)$. This assumes (implicitly) that the protected group can be seen as a treatment variable, while this has been largely debated in the literature (see e.g., for gender [Hu and Kohler-Hausman, 2020]). A proper discussion of this aspect should be provided.

W2) The overall presentation can be improved. For instance, I find the empirical evaluation in section 4 quite dense and difficult to follow.  E.g., starting the whole section from lines 310-319 can help the reader better understand the purpose of the experimental evaluation and help describe the evaluation setup (e.g., datasets, baselines, hyperparameters and metrics).

W3) The empirical evaluation can be improved. Currently, the evaluation is limited to the COMPAS and German Credit datasets, which are rather small scale. I would argue that testing CBS on larger-scale datasets such as $\texttt{folktables}$ [Ding et al., 2023] and $\texttt{WCLD}$ [Ash et al, 2023] would make the results more compelling. Moreover, I do think CBS can be exploited to audit different classifiers and their relative biases, even though such an experiment is not performed.


[Hu and Kohler-Hausman, 2019] - Hu, Lily, and Issa Kohler-Hausmann. "What's sex got to do with machine learning?." In Proceedings of the 2020 Conference on Fairness, Accountability, and Transparency, pp. 513-513. 2020.

[Ruggieri et al., 2023] - Ruggieri, Salvatore, Jose M. Alvarez, Andrea Pugnana, Laura State and Franco Turini. "Can we trust fair-AI?." In Proceedings of the AAAI Conference on Artificial Intelligence, vol. 37, no. 13, pp. 15421-15430. 2023.

[Ding et al., 2021] - Ding, Frances, Moritz Hardt, John Miller, and Ludwig Schmidt. "Retiring adult: New datasets for fair machine learning." Advances in neural information processing systems 34 (2021): 6478-6490.

[Ash et al., 2023] - Ash, Elliott, Naman Goel, Nianyun Li, Claudia Marangon, and Peiyao Sun. "WCLD: curated large dataset of criminal cases from Wisconsin circuit courts." Advances in Neural Information Processing Systems 36 (2023): 12626-12643.

**Questions:**

I have a few questions for the authors.

* q1) Could the authors clarify my doubts regarding W1)? In particular, what are the authors' considerations regarding the emergence of the Yule effect?
* q2) From Figure 3, it seems to me that the final results are heavily affected by which is the protected class specified.  E.g., if we specify the "Black defendants" as the starting protected class, the subgroup with the highest F score  (on Separation Scan for Recommendations) is "Black Male Defendants", while if we set "Male Defendants" as the starting class, the subgroup with the highest F score is "Male Asian and Hispanic Defendants". Even if I see why this occurs (in the first case, we refer to non-Black as the reference group, and in the second case, we refer to Females as the reference group), I would argue that this affects the ability of CBS to assess the intersectional biases occurring on a dataset. Can you comment on my observation further?

---

### Official Review · Reviewer_wtcB · 2024-11-01

**Soundness:** 2
**Presentation:** 1
**Contribution:** 2
**Rating:** 3
**Confidence:** 4

**Summary:**

The authors investigate the problem of intersectional bias in classification and develop a novel search method for identifying intersectional bias. The authors compare their method to other auditing methods on semi-synthetic data.

**Strengths:**

- **[Problem Importance]** The authors study an important problem.

- **[Practicality]** The proposed method can accommodate a large number of fairness definitions that prior works are not able to accommodate.

**Weaknesses:**

- **[Clarity]** Several important aspects of the paper are not articulated clearly. In particular, I found it difficult to follow the authors’ experimental design, a few examples include:
     - Section 4: What is a “row”? The authors refer to specific rows or “row $i$” without defining this term. Is the row a particular data point, or is it a row of the covariates?
     - When defining the true log-odds in their semi-sythetic data the authors say, “We use these weights to produce the true log-odds of a positive outcome $(Y_i = 1)$ for each row $i$ by a linear combination of the attribute values with these weights.”
This statement is quite vague and does not rigorously outline how the log-odds are computed. The authors denote the true log-odds as $L_i^{\text{true}}$; perhaps a definition could be given for this quantity.


- **[Narrative vs Experiments]** There is a strong disconnect between the authors' results and the discussion/motivation of the paper. For example, the authors spend substantial time going over different fairness metrics and discussing the applicability of their method to each metric. However, no experimental results are shown for any such metrics. The authors simply shift the predicted probabilities, or true probabilities, of some individuals by some value and measure whether their algorithm, or the baseline, can identify those individuals. I would have liked to see some results showing the accuracy of the authors' method as a function of subgroup unfairness under a particular metric.

- **[Synthetic Data]** Due to the way in which the synthetic data is constructed, I find it difficult to appreciate some of the authors’ results. In particular, the authors randomly select sensitive attributes among all attributes in the data and change the true labels to have a noisy linear relationship with the features. Both choices destroy the innate relationships between features, sensitive attributes, and true labels, which cause unfairness in the base datasets (e.g., COMPAS). Further, it is not clear to me why we need synthetic data in the first place. The authors are working with two datasets that are known to possess innate bias both at the group level and the subgroup level; this begs the question as to why we are not shown results comparing the authors' method to SotA methods on these datasets without any synthetic modifications.


- **[Simplistic Experiments]** In addition to the issues with synthetic data above, the authors only show experiments for two datasets and two classifier types.

- **[Evaluation Metric]** When comparing to the baseline, the authors measure the IOU of the predicted subgroups $S^*$ and the subgroups with injected bias $S_{\text{bias}}$, given on line 371. Without knowing whether or not the subgroups in $S_{\text{bias}}$ are disadvantaged (and to what degree), it is difficult to appreciate the use of this metric.

- **[Comparisons to Baselines]** When comparing their method to baselines, e.g., in Figure 1, the authors find that their method is only superior to baselines for relatively large amounts of bias. Moreover, in results such as Figure 2, it seems that the authors’ methods can achieve extremely poor accuracy depending on the type of bias present (negative or positive delta). Without apriori knowing the type of bias, it may be difficult to meaningfully apply these methods in practice. Lastly, the results in each of the aforementioned, and similar plots are difficult to interpret because we cannot understand how much bias a specific value of $\delta$ or $\mu$ corresponds to. It would be helpful to see the level of bias converted into actual fairness metrics.

**Questions:**

Could the authors please address my concerns in the above section?

---

### Official Review · Reviewer_yyee · 2024-11-09

**Soundness:** 3
**Presentation:** 3
**Contribution:** 3
**Rating:** 6
**Confidence:** 3

**Summary:**

This study develops a new statistical test for identifying bias in prediction models across four different axes based on both the probabilistic outputs and the binarized classifications. This test builds upon likelihood ratio tests developed in the spatial and subset scan statistics literature. At it's core this test is examining when the quantity of interest deviates significantly from its expectation across multiple intersectional subgroups. The test is then evaluated on a semi synthetic dataset based on COMPAS and then COMPAS itself. Demonstrating its ability to identify bias and the most significantly impacted groups.

**Strengths:**

- The method proposed is relatively simple to implement and rigorously grounded in the hypothesis testing literature
- The method is flexible for the commonly discussed fairness metrics
- The synthetic experiments are designed well to demonstrate the efficacy of the method in different scenarios and metrics

**Weaknesses:**

- The methods that are compared against seem to be quite old and I would be interested to see how they compare to newer methods in the literature (e.g. [1])
- More real world dataset studies would improve the study (e.g. folktables [2])
- The writing is verbose at times and could benefit from being more concise. This is especially true in Section 3 when describing the methods.

[1] Cherian, John J., and Emmanuel J. Candès. "Statistical inference for fairness auditing." Journal of Machine Learning Research 25.149 (2024): 1-49.
[2] https://github.com/socialfoundations/folktables

**Questions:**

1. How difficult does it appear to be to extend this framework to a multi-label setting?
2. How computationally expensive does this method get when scaling the number of groups to be evaluated over? As it stands it appears that the maximum number of intersectional groups is 4 in these experiments?

---

### Note · Authors · 2024-11-25

I have read and agree with the venue's withdrawal policy on behalf of myself and my co-authors.